# Magnon spectroscopy in the electron microscope

Demie Kepaptsoglou[1,2,3,10 ✉], José Ángel Castellanos-Reyes[4,10], Adam Kerrigan[2,3], Júlio Alves do Nascimento[2,3], Paul M. Zeiger[4], Khalil El hajraoui[1,2], Juan Carlos Idrobo[5,6], Budhika G. Mendis[7], Anders Bergman[4], Vlado K. Lazarov[2,3], Ján Rusz[4 ✉] & Quentin M. Ramasse[1,8,9 ✉]

The miniaturization of transistors is approaching its limits owing to challenges in heat management and information transfer speed[1]. To overcome these obstacles, emerging technologies such as spintronics[2] are being developed, which make use of the electron's spin as well as its charge. Local phenomena at interfaces or structural defects will greatly influence the efficiency of spin-based devices, making the ability to study spin-wave propagation at the nanoscale and atomic scale a key challenge[3,4]. The development of high-spatial-resolution tools to investigate spin waves, also called magnons, at relevant length scales is thus essential to understand how their properties are affected by local features. Here we detect bulk THz magnons at the nanoscale using scanning transmission electron microscopy (STEM). By using high-resolution electron energy-loss spectroscopy with hybrid-pixel electron detectors, we overcome the challenges posed by weak signals to map THz magnon excitations in a thin NiO nanocrystal. Advanced inelastic electron scattering simulations corroborate our findings. These results open new avenues for detecting magnons and exploring their dispersions and their modifications arising from nanoscale structural or chemical defects. This marks a milestone in magnonics and presents exciting opportunities for the development of spintronic devices.

The use of spin currents in information transistors is predicted to offer non-volatility, faster data processing, higher integration densities and lower power consumption[5-7], thanks to degrees of freedom emerging in quantum materials that exhibit unique spin-dependent properties, including topologically protected spin states[5]. At the same time, recent developments in antiferromagnetic spintronics have demonstrated that antiferromagnetic materials, such as NiO, provide a promising platform for spin injection and transport in the THz domain[8] and spin-torque control[9]. Magnons, the collective excitations of the spin lattice in ferromagnets and antiferromagnets, which can be visualized semiclassically as a wave of precessing magnetic moments[10], are becoming a cornerstone of quantum technology[11] through proposed spin-current-based device architectures. Of crucial importance are localized phenomena at the nanoscale and atomic scale, such as the scattering of spin waves at heterointerfaces and structural defects in materials, which can affect spin injection, spin-wave transmission, spin-torque switching and spin-to-charge conversion. As a result, the ability to achieve (sub-)nanometre-resolution magnon detection is considered one of the main challenges in the field of magnonics[3,4].

Electrons as a probe for magnon excitations are commonly used through surface scattering of low-energy electrons in reflection high-resolution electron energy-loss spectroscopy (HREELS), using either spin-polarized or non-polarized electron sources[10,12-14]. Although HREELS can examine the energy–momentum dispersion of magnons with high energy resolution, it is limited to the study of surface excitations of ultrathin films over large length scales owing to limitations in the scattering cross-sections, spatial resolution and penetration depth intrinsic to the technique. Similarly, other experimental approaches widely used to study magnons at high energy and momentum resolutions, such as inelastic neutron scattering[15], time-resolved Kerr microscopy[16] or Brillouin light scattering[17], are also fundamentally limited to spatial resolutions of hundreds of nanometres and often to large sample volumes. Consequently, magnon information from nanometre-sized features, such as defects and buried interfaces, is not accessible. New approaches in vector magnetometry using nitrogen-vacancy (NV⁻) centres sensing have recently shown great promise in mapping surface magnetic textures and detecting magnons at the nanometre-scale with high sensitivity[18,19], although the fabrication of NV⁻-point-defect sensors remains challenging.

Since its first demonstration[20,21], meV-level (vibrational) electron energy loss spectroscopy (EELS) in a scanning transmission electron microscope has been developing at a swift pace. Several key experimental milestones have been achieved: the detection of atomic-level contrast in vibrational signals[22], the spectral signature of individual

[1]SuperSTEM Laboratory, Sci-Tech Daresbury Campus, Daresbury, UK. [2]School of Physics, Engineering and Technology, University of York, Heslington, UK. [3]York JEOL Nanocentre, University of York, Heslington, UK. [4]Department of Physics and Astronomy, Uppsala University, Uppsala, Sweden. [5]Materials Science and Engineering Department, University of Washington, Seattle, WA, USA. [6]Physical and Computational Sciences Directorate, Pacific Northwest National Laboratory, Richland, WA, USA. [7]Department of Physics, Durham University, Durham, UK. [8]School of Chemical and Process Engineering, University of Leeds, Leeds, UK. [9]School of Physics and Astronomy, University of Leeds, Leeds, UK. [10]These authors contributed equally: Demie Kepaptsoglou, José Ángel Castellanos-Reyes. ✉e-mail: dmkepap@superstem.org; jan.rusz@physics.uu.se; qmramasse@superstem.org

impurity atoms[23], spatially resolved measurements on point and line defects in crystalline materials[24,25], as well as momentum-resolved measurements using nanoscale beams[26,27]. With energy losses owing to magnon excitations occupying the same spectral window as phonons, ranging from a few to a few hundred meV in solid-state materials[10,12–14], the promise of detecting magnons in an electron microscope is exciting from both fundamental research and applications points of view.

Recent theoretical studies on inelastic magnon scattering in an electron microscope confirmed the detectability of magnons as diffuse inelastic scattering and demonstrated the tantalizing prospect of obtaining atomically localized magnon information[28–30]. This exploratory work also highlighted experimental challenges such as the separation of phonon from magnon diffuse scattering, as the latter is predicted to be several orders of magnitude weaker than the former and yet occupy a similar energy-loss span. Here we tackle the challenges of detecting magnons at the nanoscale using high-resolution EELS in the scanning transmission electron microscope and we present the first direct detection of magnons with STEM-EELS. Furthermore, we demonstrate that, at the interface between a NiO thin film and a non-magnetic substrate, the magnon signal is exclusively confined within the film, confirming that magnons can be mapped with nanometre spatial resolution. We show that, although challenging, the detection of the inherently weak magnon signal is possible, thanks in part to the dynamic range of hybrid-pixel electron detectors[31,32]. The experiments are supported by state-of-the-art numerical simulations of electron scattering[33], underpinned by atomistic spin dynamics (ASD) simulations[34].

The primary challenge in achieving magnon EELS is that the energy ranges of phonon and magnon losses overlap, with the weaker magnon signal likely to be overshadowed by the inherently stronger lattice vibration modes. However, these types of loss follow different dispersion relations[35] and, thus, should be differentiable in momentum-resolved experiments, given a suitable choice of material whose magnon and phonon branches are sufficiently separated in momentum and energy. For this purpose, we have selected NiO as a model system; as well as being of interest for spin-transfer-based devices[36], its dispersion relations of phonon and magnon modes have been shown to meet our requirements of momentum and energy separation in the THz range[37,38].

A schematic representation of the experimental geometry used for the momentum-resolved experiments is presented in Fig. 1a. The instrument's electron optics are adjusted to a low convergence angle to form a diffraction-limited nanometre-sized electron probe, while achieving sufficient momentum resolution (Methods). The electron probe is kept stationary on a region of interest of the NiO crystal, across which the zone-axis orientation is perfectly maintained, estimated to extend no further than a few nanometres from the nominal probe position. The stability of the microscope sample stage, with typical drift measured of less than 0.5 nm per hour, enables hours-long acquisitions on a nanometre-sized area of the NiO crystal (Methods), a timescale still far shorter than necessary for magnon spectroscopy experiments using, for example, inelastic neutron scattering, for which days of integration from bulk samples can be required. A narrow rectangular (slot) collection aperture for EELS is used for the angle-resolved measurements[39]. The slot aperture is aligned to select a row of systematic Bragg reflections (Fig. 1b) and to produce two-dimensional intensity maps of energy loss (or frequency) $\omega$ versus momentum transfer **q**.

Figure 2 shows examples of such measurements acquired along the 220 and 002 rows of systematic Bragg reflections in NiO (corresponding to the $\Gamma \to M$ and $\Gamma \to X$ **q** paths, respectively). Figure 2a,b shows the as-acquired intensity maps of energy versus momentum transfer ($\omega$–**q** maps). The $\omega$–**q** maps show two intense bands dispersing around 30 and 50 meV, which correspond to the NiO longitudinal-acoustic (LA) and longitudinal-optical (LO) phonon branches, respectively, in agreement with previous experimental[37] and theoretical work[38,40]. These are labelled on Extended Data Fig. 2, in which gain LA phonon

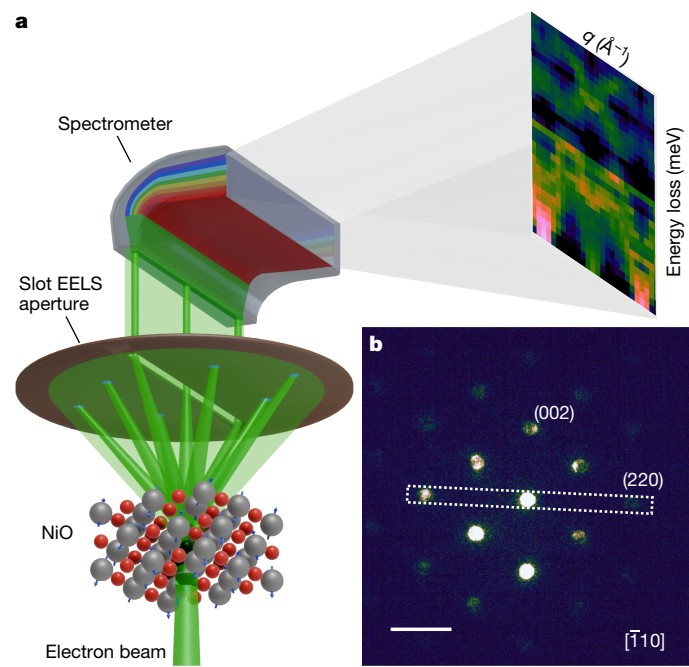

**Fig. 1 | Experimental geometry of momentum-resolved EELS. a**, Schematic representation of the geometry of $\omega$–**q** vibrational EELS measurements using a rectangular slot collection aperture. **b**, Experimental diffraction pattern along the NiO [$\bar{1}$10] zone axis at a 2.25 mrad convergence angle, with the monochromating slit inserted, showing the orientation of the slot EELS collection aperture along the 220 row of systematic Bragg reflections in diffraction space. Scale bar, 2.5 Å$^{-1}$, 20 mrad.

branches are also visible (higher energy-gain branches are outside the recorded energy window).

Because the intensity of magnon EELS is expected to be much lower than that of phonons[28,30], and given that magnon modes in NiO occur at higher energy losses compared with phonons, scaling the data by the square of the energy loss (intensity $\times E^2$) provides a useful means to enhance the visibility of weaker features above the decaying zero-loss-peak tail in the meV range, while avoiding possible errors in background fitting; see Supplementary Note 1. The scaled but otherwise unprocessed $\omega$–**q** maps, presented in Supplementary Fig. 1b,e, readily show further spectral bands above 80 meV along $\Gamma \to M$ and $\Gamma \to X$, which are present but hard to discern in the non-scaled data.

Subtracting the background from the tail of the phonon signal using a first-order log-polynomial model offers a clear illustration of the dispersion of the two spectral bands above 80 meV, shown in Fig. 2c,d for the $\Gamma \to M$ and $\Gamma \to X$ directions (with signal-to-noise ratios (SNRs) of 22 and 59, respectively; see Supplementary Note 3 for details on SNR estimation). Supplementary Note 2 explores the robustness of different background models.

In Fig. 2c, two lobes of spectral intensity are visible on either side of the M point, with a maximum peak at 100 meV. The intensity of the lobes is asymmetric around the M point, with a stronger signal between $\Gamma$ and M compared with the lobe between M and $\Gamma'$. The intensity tends to 0 towards each of the Brillouin zone vertices. The same asymmetry is also evident in the 002 data (Fig. 2d) but the separation between lobes is less pronounced along the shorter $\Gamma \to X$ distances owing to limited momentum resolution and spectral smearing caused by averaging. Supplementary Fig. 3 shows how better branch separation can be achieved with less averaging at the cost of higher noise and lower SNR, as discussed in Supplementary Note 3.

The presence of these bands in the non-scaled data is confirmed in Fig. 2e by the integrated intensity profiles of the $\omega$–**q** map over a narrow momentum window ($\Delta q = 0.22$ Å$^{-1}$, to avoid spectral broadening

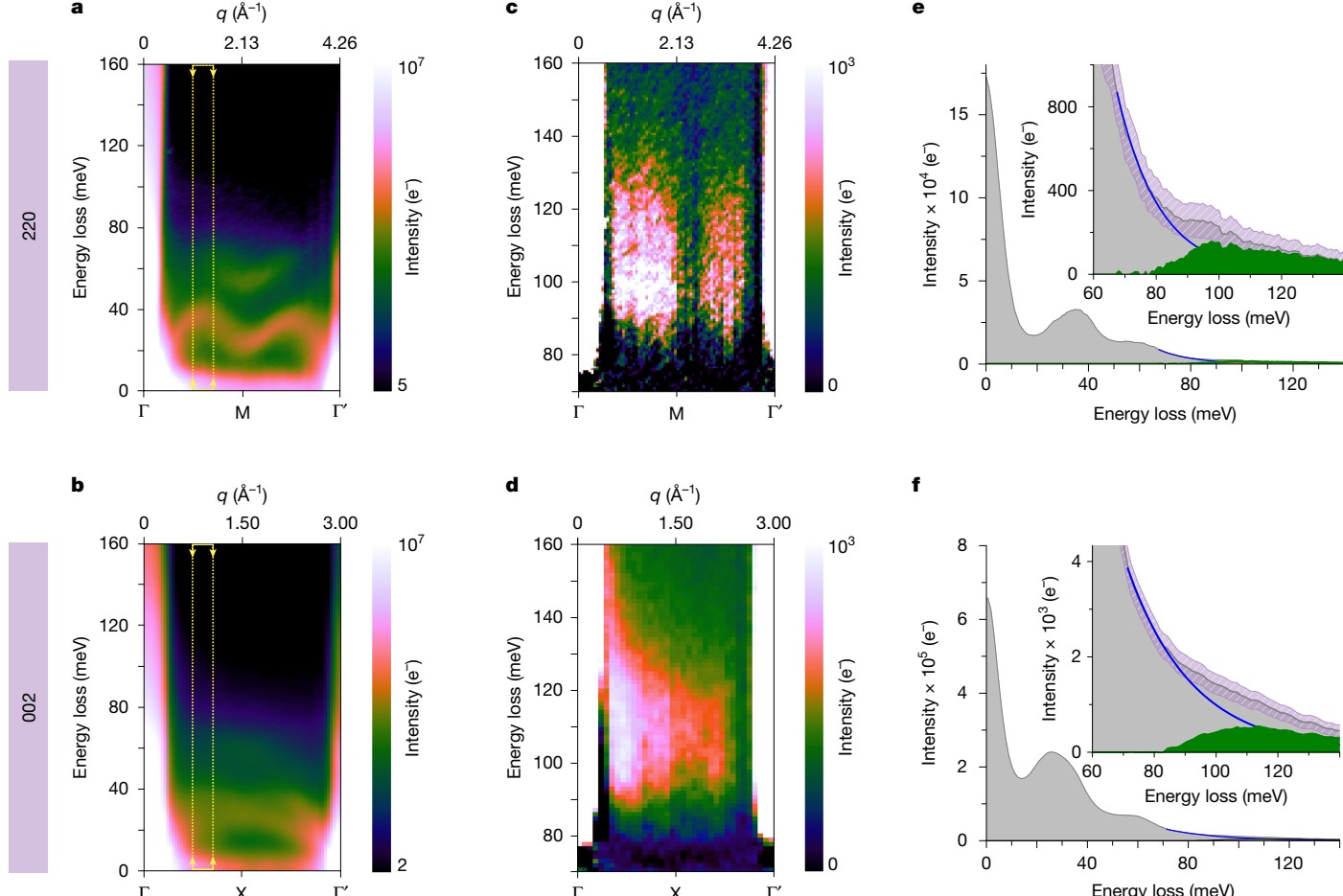

**Fig. 2 | Momentum-resolved vibrational EELS measurements of NiO. a,b**, As-acquired $\omega$–**q** maps along the 220 and 002 rows of reflections, respectively, showing the dispersion of the NiO LA and LO phonon branches. For clarity, the intensity of the maps (calibrated in electrons, e⁻) is shown on a logarithmic colour scale. **c,d**, Background-subtracted $\omega$–**q** maps showing the dispersion of the magnon bands at about 100 meV. **e,f**, Integrated spectra at the momentum positions marked by arrows in panels **a** and **b**. Insets,

background-subtracted spectra (green-shaded area), obtained by removing a first-order log-polynomial model (blue line) from the raw signal (grey-shaded area). Pink-shaded areas illustrate the error bars as confidence bands at a +/−5$\sigma$ level ($\sigma$ is the standard deviation, calculated by assuming that the magnon scattering and noise populations are Poisson distributed; see Supplementary Note 3).

through momentum averaging) at $q = 1.24$ Å⁻¹, the value at which the intensity is maximum along $\Gamma \to$ M, marked by the yellow arrows and dashed lines in Fig. 2a. This highlights the shape of the peak, with a rising edge from 80 meV reaching a maximum at about 100 meV, before a weaker feature extending up to 120 meV. Similarly, in the integrated intensity profile of the $\Gamma \to$ X **q** path in Fig. 2f, the band at 100 meV is observed at $q = 0.97$ Å⁻¹ (over a $\Delta q = 0.2$ Å⁻¹ window; Fig. 2d), comparatively further away from the $\Gamma$ point than the band observed in the $\Gamma \to$ M direction. This signal is unambiguously above the experimental error, as illustrated in Fig. 2e,f by confidence bands (pink-shaded areas) at a 5$\sigma$ confidence level ($\sigma$ is the standard deviation, calculated by assuming that the magnon scattering and noise populations are Poisson distributed; see Supplementary Note 3).

The observed spectral bands emerge in the same energy–momentum space in which magnon modes are expected for NiO. The magnon density of states is known to shift to higher energies with decreasing temperature[41], so the acquisition temperature difference explains a roughly 20 meV blueshift between EELS (room temperature) and neutron experiments (10 K in ref. 37), a conclusion borne out by simulations discussed below. Furthermore, the asymmetry of detected bands along both **q** paths, including their appearance most prominently above background further away from $\Gamma$ in the $\Gamma \to$ X direction than along the $\Gamma \to$ M direction, is consistent with neutron-scattering

experiments[37,41]. Here the asymmetry is also probably the result, in part, of the lower intensity away from the direct beam.

Owing to the overlap of the elastic (or zero-loss peak (ZLP)), phonon and magnon signals along **q**, it is difficult to quantify the absolute intensity of the observed bands we attribute to magnons. Nevertheless, an estimate can be given by the integral of the signal within the $\Gamma \to$ M $\to \Gamma'$ window (excluding $\Gamma$ points); after background subtraction (Fig. 2c), the integrated magnon intensity is estimated as roughly $8.5 \times 10^4$ e⁻. For comparison, the corresponding integrated phonon intensity (for all branches) over the same $\Delta q$ is three orders of magnitude higher, about $2.0 \times 10^7$ e⁻, whereas the total integrated intensity across the slot aperture, assumed to be representative of the total beam intensity impinging on the sample, was $5.2 \times 10^9$ e⁻, in agreement with the expected relative intensities of the magnon and total scattered EELS signals[28]. A similar intensity analysis holds for the $\Gamma \to$ X data.

To support our experimental findings, we have performed first-principles calculations of momentum-resolved phonon and magnon EELS using parameters that reflect the experimental conditions, in particular the sample temperature, magnetic environment and the electron-optical parameters of the microscope (Methods). Figure 3 summarizes the results of the simulations for both experimental **q** paths. We observe phonon EELS bands reaching up to about 70 meV for both **q** paths, with a small gap around 40 meV (Fig. 3a,c). This

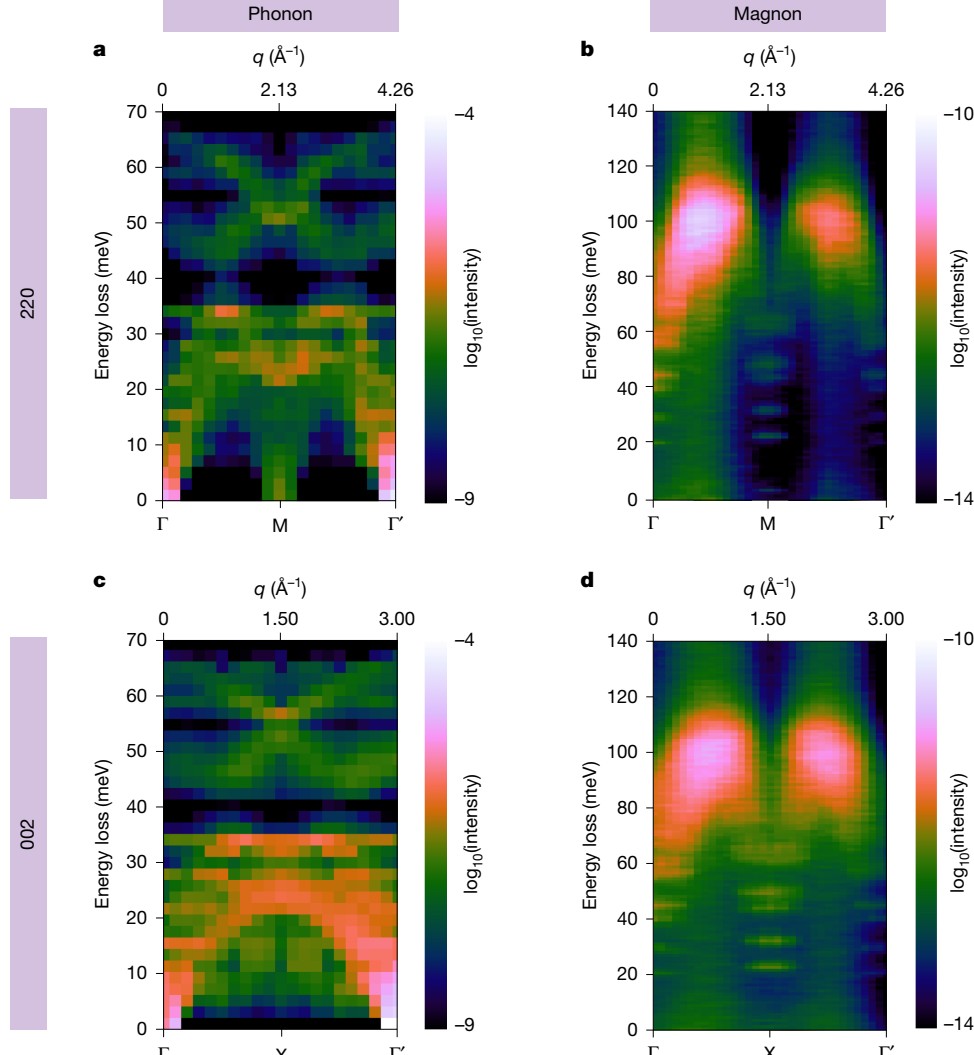

**Fig. 3 | Calculated vibrational and magnon EELS of NiO. a–d**, Simulated momentum-resolved EELS (dispersion curves) of phonon (**a**,**c**) and magnon (**b**,**d**) excitations, along the Γ → M and Γ → X **q** paths of the Brillouin zone for NiO, respectively. Experimental parameters such as sample temperature, environmental magnetic field inside the microscope and electron-optical parameters are considered (Methods).

matches the two dominant features observed in the experimental momentum-resolved phonon EELS datasets, albeit with a slightly higher energy-loss gap at 50 meV. Owing to the long acquisition times required to reveal the emergence of the magnon bands, and the chosen balance between momentum and spatial resolutions, some spectral smearing obscures the finer details of the experimental phonon dispersion. For completeness, a better-resolved $\omega$−**q** map is presented in Extended Data Fig. 2 comparing very favourably with the calculated phonon dispersion in Fig. 3a (but in which the magnon signal is fainter as a result of shorter integration), and data accumulation is discussed in Supplementary Note 3. The rich pattern observed in phonon simulations contrasts with magnon simulations, which show negligible fractional scattering intensities at energy losses below 60 meV, with two broad-yet-well-isolated peaks at energy losses between 80 and 120 meV (Fig. 3b,d).

The theoretical calculations closely match the features emerging above the phonon EELS background observed in experiments (Fig. 2). The maximum simulated magnon EELS intensity appears around 100 meV at room temperature, as observed experimentally, whereas simulations of the magnon dispersion at 10 K (Extended Data Fig. 3) predict an energy-loss peak close to 120 meV, as expected from neutron experiments at this temperature. The measured asymmetry along the

momentum axis, also present in neutron-scattering data, is well reproduced, showing a considerably higher intensity for the magnon EELS peak closer to the Γ point in the centre of the diffraction plane when compared with the peak near the 220 or 002 Γ′ points, respectively. This is intrinsic to the magnon EELS scattering strength, although, as discussed above, it is exacerbated experimentally owing to the lower intensity away from the direct STEM-EELS beam. Similarly, along the momentum direction, the calculated magnon intensity maximum is comparatively closer to the X point than to the M point, as in the experiments. Furthermore, the intensity peak shape in the energy-loss direction matches the experimental observations, with an extended spectral tail towards 120 meV in the Γ → M calculations, compared with a slightly more rounded peak centred at 100 meV in the Γ → X case. Theoretical and experimental integrated spectra and dispersions are shown side by side in Extended Data Fig. 4 and a broadened version of the simulations to mimic the finite experimental resolution is shown in Supplementary Fig. 4. These illustrate the excellent match and confirm the interpretation of the experimental intensities around 100 meV as being the spectral signature of scattering arising from magnon excitations in the nanometre-sized NiO crystal.

A key advantage of STEM-EELS is its unique ability to examine spectral signals at high spatial resolution. Spatially resolved measurements

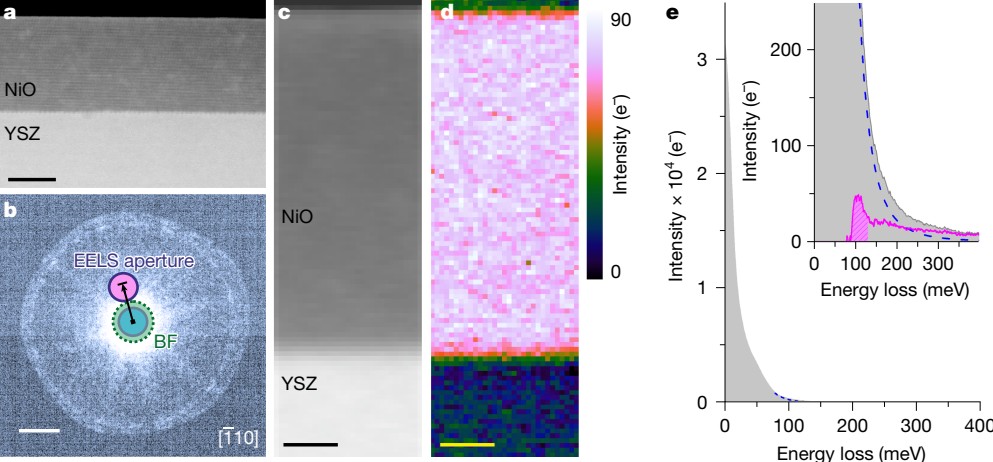

**Fig. 4 | Spatially resolved magnon EELS measurements across a NiO thin film. a**, High-angle annular-dark-field image of a 30-nm NiO thin film grown on a YSZ substrate. **b**, Experimental diffraction pattern along the NiO [$\bar{1}$10] zone axis at a 31 mrad semi-convergence angle, with the monochromating slit inserted, showing the off-axis displacement of the round EELS collection aperture along the [002] direction (marked by the black arrow). **c**, Asymmetric (displaced) annular-dark-field image acquired during EELS measurements. **d**, Integrated intensity map of the magnon peak over the energy window indicated by a shaded area, after background subtraction, as illustrated in **e** with the signal integrated over the whole NiO film area (the background-subtracted version is presented, pink curve, inset). Scale bars, 15 nm (**a**); 7.7 Å$^{-1}$, 60 mrad (**b**); 5 nm (**c**,**d**).

across a 30-nm-thick film of NiO grown on top of an yttrium-stabilized zirconia (YSZ) substrate demonstrate the potential of this technique for mapping magnons at the nanometre scale. Figure 4 shows a dark-field EELS experiment[22], in which an angstrom-sized electron probe is rastered across the sample, the EELS signal being collected by a circular aperture displaced off the optical axis (Methods). This geometry was suggested in earlier work to be favourable for spatially resolved magnon detection[28]. An averaged spectrum shows a well-defined magnon peak at 100 meV (Fig. 4e), whereas—notably—the integrated map (Fig. 4d) shows that the signal is confined within the NiO film, disappearing completely within a sub-nanometre distance of the interface with the (non-magnetic) YSZ substrate and into the hole above the film surface. Spatially resolved spectral variations are explored in Supplementary Note 6.

Inelastic magnon simulations for a thin slab of NiO are consistent with the experiments and predict a smaller intensity of the magnon signal at the edges of the slab compared with its centre, with a broader, more featureless post-peak tail (Supplementary Figs. 7 and 8). The thorough interpretation of signal differences near heterointerfaces would require the development of appropriate modelling frameworks and the observed drop in signal intensity observed near the interface with YSZ may have other causes (for example, beam propagation). Nevertheless, this experiment demonstrates unambiguously the nanometre-scale mapping of magnons in an electron microscope.

## Discussion and conclusions

Our observation of magnon excitations in STEM-EELS arrives a decade after the milestone report of the detection of phonons in an electron microscope[20]. We may reasonably expect further developments of magnon EELS to follow a similar trajectory and pace as its phonon counterpart, with a blueprint for studies of magnon dispersions and their nanoscale modifications in the vicinity of surfaces, interfaces or defects. These will create a radical new way of studying magnetism and spin-wave excitations at the length scales relevant for device fabrication, using widely applied numerical analysis techniques, for example, to extract exchange interaction parameters, using spatially resolved experimental data. In this context, we emphasize that, as well as the spatially resolved magnon map in a thin film (Fig. 4), the $\omega$−**q** measurements were performed with a probe diameter smaller than

2 nm, with a 10-nm-wide region of interest at most, across which the NiO single crystal was perfectly on axis to enable a careful examination of the chosen diffraction directions, as illustrated by images of the area used for the experiments (Extended Data Fig. 1). This combination of high spatial resolution with the flexibility to optimize the acquisition geometry for dispersion measurements when sample size, heterogeneity or spatial selectivity are important is unique to STEM[26]. Although momentum resolution is limited in the nanoprobe regime, balancing momentum and spatial resolutions will be necessary to understand the underlying scattering physics and spin-wave propagation in nanoscale objects.

The much lower cross-section for scattering of fast electrons from magnons compared with phonons[28,29,33] remains a challenge. In our experiments, calculations suggest that we can expect to detect one electron scattered by magnons only every 1–2 s, whereas there can be thousands of phonon-scattered electrons every second under equivalent settings. This may limit the ability to differentiate magnon dispersions from phonon branches present within the same region of $\omega$−**q** space, placing constraints on the choice of materials systems for which the direct observation of magnon peaks will be possible in the immediate future.

However, several systems of interest for magnonics, including metals[28,35] and oxides[42], present favourable separation between phonons and magnon branches, with relatively high magnon scattering cross-sections. There are therefore many materials to explore as the technique develops, for which low detection efficiency and spectral overlaps are not prohibitive.

Furthermore, alternative strategies can be deployed to enable a more effective, albeit less direct, detection of magnons in STEM-EELS in cases for which magnons and phonon branches are close. The interaction of magnons with other excitations in the same energy range can give rise to other quasiparticles such as magnon polarons (the hybridized state between phonons and magnons), which offer less challenging spectroscopic fingerprints. These interactions are characterized by modifications of the dispersion behaviour (for example, the appearance of spectral band anticrossings)[43,44]. Preliminary reports suggest that this strategy may be promising[45,46]. Similarly, the different dependence of magnons on external sample stimuli (for example, temperature) compared with phonons may provide an efficient means to disentangle the magnon signal from higher-intensity features[30].

Finally, continuing technological and methodological developments in STEM-EELS are opening experimental avenues. Electron-counting detectors enable measurements hampered only by Poisson noise[31]. Although comparatively shorter experiments are an advantage for the STEM-EELS approach, the exceptional stability of STEM instruments with low sample drift, operating at reduced acceleration voltages minimizing sample damage combined with advanced data-acquisition procedures such as multiframe recording[47], make it possible, in principle, to substantially increase acquisition times, approaching day-long timescales that are common in techniques such as angle-resolved photoemission[48] or inelastic neutron scattering, while retaining nanoscale or atomic-scale spatial resolutions. Longer acquisition times will provide an efficient mitigation strategy to circumvent the low magnon EELS signal, leading to higher-fidelity detection at high spatial resolution. With the availability of tools such as variable-field pole pieces[49] and liquid-helium-temperature stages[50], along with newer-generations monochromators, future magnon STEM-EELS experiments will gain new controls over magnetic field and extended temperature ranges. These can be used to suppress magnon or phonon modes or to enhance energy separation, marking a new era for magnonics studies at the nanometre scale.

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

## Methods

### Materials and specimen preparation

Electron microscopy specimens were prepared conventionally from a commercially available NiO single-crystal substrate (Surface Preparation Laboratory) by crushing (pestle and mortar), dispersing in chloroform and drop casting on standard lacey carbon support grids. Further specimens were prepared by focused ion beam, using a Hitachi NX5000 Ethos triple-beam instrument, from a 30-nm-thick NiO thin film grown on a YSZ substrate by molecular-beam epitaxy. Sample thickness was estimated using the standard log-ratio method from low-loss EELS spectra to be approximately 30–40 nm in the areas used for all of the data presented here. Grains with a suitable zone-axis orientation were specifically chosen so that their diffraction patterns aligned close to the main axis of the slot aperture for momentum-resolved acquisitions. Small rotational adjustments were carried out using the microscope's projector lens system, to ensure perfect alignment of the row of diffraction spots to the slot axis. When this was not possible, the sample was removed from the microscope and rotated to match the measured rotation of pre-screened grains of interest. The effective regions of interest across which the crystal areas were both thin enough, overhanging any lacey carbon support material and perfectly on zone axis (judged from the symmetry of the observed intensities in the diffraction pattern) were estimated to be no larger than 10 nm across in most cases.

### High-resolution EELS and data processing

High-resolution EELS measurements were performed in a Nion Ultra-STEM 100MC 'HERMES' aberration-corrected dedicated scanning transmission electron microscope. The instrument is equipped with a Nion-designed high-resolution ground-potential monochromator, a Nion Iris high-energy-resolution energy-loss spectrometer and a Dectris ELA hybrid-pixel direct electron detector optimized for EELS at low acceleration voltages[51]. The instrument was operated at 60 kV acceleration voltage to minimize sample damage and take advantage of higher inelastic cross-section for EELS experiments[52], with the electron optics adjusted to achieve a 2.25 mrad convergence angle corresponding to a diffraction-limited electron probe approximately 1.3 nm in diameter (Extended Data Fig. 1). Although smaller convergence angles can be achieved, this choice offered an adequate balance between probe size, momentum resolution and electron optics stability. The nominal energy resolution was 7.2 meV, set by the position of the monochromator energy-selection slit and estimated by the full width at half maximum (FWHM) of the ZLP in vacuum (Extended Data Fig. 2). Some energy broadening occurs with the beam traversing the sample and with data accumulation (see Supplementary Note 3). The resulting probe current after monochromation and with the optical parameters chosen to ensure the highest achievable energy resolution was about 1 pA. For the momentum-resolved experiments, a rectangular (slot) aperture (0.125 × 2 mm) was used, with the projector optics adjusted to match the full angular size of the beam to the width of the slit. The momentum resolution is dictated by the combined effect of the convergence and collection apertures used. This is often described using an 'effective' collection angle, the quadratic sum of the convergence half-angle $\alpha$ and collection half-angle $\beta$ (ref. 53). To be exact, the momentum resolution can be expressed as $k\sqrt{\sin^2\alpha + \sin^2\beta}$, in which $k$ is the electron beam wavenumber and $\alpha$ and $\beta$ are the convergence and collection semiangles, respectively. In the case of a rectangular momentum-selecting EELS slit, which is here chosen to have a width in the energy-dispersive direction that matches the beam convergence, it is easy to see that the momentum resolution can, for simplicity, be considered to be limited by the size of the diffraction spots on the EELS camera (that is, the full beam convergence angle). Defining the electron wavenumber as $2\pi/\lambda$ for a wavelength $\lambda = 0.0487$ Å at 60 kV acceleration voltage, this yields a momentum resolution of $\Delta q = 0.4$ Å$^{-1}$, consistent with previous nanoscale momentum-resolved experiments using

a similar set of parameters[26]. The slot aspect ratio allows an angular range in the momentum direction of 50 mrad. The pixel size along the momentum direction was 0.057 Å$^{-1}$ per pixel and 0.088 Å$^{-1}$ per pixel for the $\Gamma \rightarrow$ M and $\Gamma \rightarrow$ X directions, respectively. The spectrometer dispersion was set to 1.0 or 0.5 meV per channel (002 and 220 acquisitions, respectively). Each $\omega$–$\mathbf{q}$ dataset was a multiframe acquisition of the whole two-dimensional extent of the spectrometer camera. A typical dataset comprises 15,000 frames, using an exposure time of 75 ms per frame, resulting in an acquisition time of 22 min per single dataset (limited only by the data-export capabilities of the microscope operating software). The multiframe stacks were subsequently assessed for energy drift, aligned using rigid image registration and integrated. The final data presented here are sums of several such integrated datasets acquired consecutively (necessary as explained above owing to data-size-handling limitations) from the same specimen area, on the same day and under identical experimental conditions. No other post-processing was applied. The impact of this averaging on practical energy and momentum resolution is explored in Supplementary Note 3.

Specifically, the dataset in Fig. 2a is the sum of 90,000 individual camera frames, acquired along the [$\bar{1}$10] zone axis of NiO (Fig. 1b) with a dwell time of 75 ms, corresponding to a total of 2 h acquisition time. Owing to the integration and averaging of several datasets, some spectral smearing results in an effective energy resolution for the experiment of 9.2 meV, measured by the FWHM of the ZLP in the final integrated dataset. The dataset in Fig. 2b is the sum of 60,000 individual camera frames, acquired along the [100] zone axis of NiO (Extended Data Fig. 1) with a dwell time of 75 ms, corresponding to a total of 1.2 h acquisition time. The effective energy resolution of the experiment was 11 meV, measured by the FWHM of the ZLP in the integrated dataset. The smaller number of frames required to obtain comparable SNR in the case of the 002 data (Fig. 2b) is because of a tip change occurring between the two sets of experiments, resulting in higher probe currents available at otherwise similar electron optical settings.

The reproducibility of the results was thoroughly assessed through the acquisition of numerous independent datasets, across several days of experiments and on various areas of the samples, all of which exhibited the same spectral features. For completeness, an extra dataset is presented in Supplementary Fig. 5. The likelihood of spurious carbon-related vibrational modes affecting the results (owing to adventitious carbon build-up or neighbouring support film) was fully excluded, as spectra recorded on an area of protective carbon on the sample prepared by focused ion beam showed an entirely different energy loss and spectral fingerprint, as illustrated in Supplementary Fig. 6 and associated discussion.

Spatially resolved high-resolution EELS measurements were performed in the dark-field (off-axis) geometry[20], using a 31 mrad convergence semiangle corresponding to about 1 Å electron probe diameter (Fig. 4). The EELS collection semiangle was 22 mrad, with the spectrometer entrance aperture displaced by 55 mrad (or 7 Å$^{-1}$ momentum transfer) along the [002] direction in momentum space. The energy resolution was 9 meV, as estimated by the FWHM of the ZLP in vacuum and the spectrometer dispersion was set to 1.0 meV per channel, with a camera dwell time of 15 ms per pixel. The data presented in Fig. 4c–e are the sum of eight multipass spectrum images (each containing ten spectrum image frames, for a total of 80 frames) with the electron beam rastered over a 15 × 40-nm area. Each spectrum image was aligned for ZLP and spatial drift before summing the frames[48]. For the map presented in Fig. 4d, the spectral data were denoised using principal component analysis, as implemented in Gatan's GMS 3.6 software suite. The intensity map of the magnon signal was generated from the integrated spectral intensity range of 80–130 meV after background subtraction using a first-order log-polynomial background to model the decaying intensity tail underlying the preceding phonon signal. A detailed analysis of observed spatial variations of the spectral signal

is provided in Supplementary Note 6 and Supplementary Fig. 6, including a comparison of spectral signatures of the NiO thin film, the YSZ substrate and the carbon-based protective sample cap, confirming the distinct and unique fingerprint of the magnon signal.

## Theoretical calculations

The study of phonon and magnon excitations through EELS requires computational methods that accurately capture complex physical effects, such as multiple scattering and dynamical diffraction, while addressing the specific challenges posed by different materials systems. To meet these requirements, we use two distinct approaches: the time autocorrelation of auxiliary wavefunctions (TACAW) and frequency-resolved frozen phonon multislice (FRFPMS) methods. These are described below with an overview of their physics, implementation and application in this work.

The TACAW method[33] is a new and versatile framework for simulating angle-resolved EELS of both phonons and magnons. It works by Fourier transforming the temperature-dependent time autocorrelation of the auxiliary electron beam wavefunction. For phonons, TACAW relies on frozen phonon multislice simulations[54] using molecular dynamics to generate atomic displacement snapshots. For magnons, it uses frozen magnon multislice simulations[28], incorporating ASD to model magnetic excitations. This method captures several scattering, thermal and dynamical diffraction effects, enabling the resolution of both energy-loss and energy-gain processes.

The FRFPMS method[55], on the other hand, focuses specifically on phonon excitations. In this work, it uses snapshots of atomic displacements derived from density functional theory (DFT) simulations to represent vibrational modes. These snapshots are grouped into frequency bins for detailed spectral decomposition. Although TACAW could not be applied to phonons in this case because of the absence of an accurate molecular dynamics potential for NiO, FRFPMS can be readily applied to snapshots with atomic displacements generated using phonon dispersions calculated by DFT.

Below, we detail the methodologies used, beginning with the TACAW-based magnon EELS simulations, followed by the FRFPMS phonon EELS calculations.

**Magnon EELS simulations.** The numerical simulations were performed with the TACAW method[33]. Specifically, the momentum-resolved EELS signals were computed as the time-to-energy Fourier transform of the temperature-dependent time autocorrelation of the electron beam wavefunction obtained from the magnetic (Pauli) multislice method[56]. ASD simulations in UppASD[34] were conducted on a 16 × 16 × 96 supercell (of dimensions 6.672 × 6.672 × 40.032 nm) of NiO cubic unit cells, using the experimental parameters reported in ref. 57. For the spin-Hamiltonian parameters, we used the values from the bottom row of Table III in ref. 57. Given that we used a cubic cell, the nearest-neighbour exchange interaction $J_1$ was set as the average of $J_1^+$ and $J_1^-$ from the same row. To account for the effect of the microscope's objective lens on the sample environment, a 1.5 T static external magnetic field oriented along the [001] direction was included. Oxygen atoms were excluded from the ASD simulations owing to their negligible magnetic moment. Using a Gilbert damping $\alpha = 5 \times 10^{-4}$ and a time step of 0.1 fs, a thermalization phase of 70,000 steps was followed by a 7.813 ps trajectory at 300 K for generating snapshots every 13 fs, enabling the exploration of magnon frequencies up to 159 meV. Note that, from ASD simulations, we determined that the used parameters of the spin-Hamiltonian predict a Néel temperature of 304 K, instead of the known experimental value of 523 K. Therefore, in our calculations, we used an effective temperature $T_e = 174.4$ K, as 300 K = $T_e$ (523/304). In total, 6,002 snapshots were generated and the magnon EELS signal was obtained as the average over 115 sets of 301 consecutive snapshots mutually offset by 50 snapshots. Each set spanned 3.913 ps, providing an energy resolution of 1.06 meV.

The electron beam exit wavefunctions were computed using the Pauli multislice method on a numerical grid of 1,344 × 1,344 points with 4,032 slices across the thickness of the NiO supercell, including the oxygen atoms (with a zero magnetic moment). A 60 kV electron probe with a 2.25 mrad convergence semiangle, propagating along the [001] direction, was used. We have adopted the parametrized magnetic fields and vector potentials developed in ref. 58 The Debye–Waller factor (from Table S.V in the supplemental material of ref. 57) and the absorptive optical potential (see Appendix B of ref. 30) were included to simulate, in a first approximation, the effect of phonon excitations on elastic scattering.

**Phonon EELS calculations.** We have used FRFPMS[56] with 34 frequency bins ranging from 0 THz up to 17 THz in 0.5-THz-wide intervals. Within each of the frequency bins, we have averaged over 128 structure snapshots. The supercell size for FRFPMS simulations was 10 × 10 × 98 cubic Bravais unit cells of NiO. For phonon EELS, a conventional multislice algorithm was used, as implemented in the Dr. Probe code[59], using a numerical grid of 840 × 840 × 784 pixels. Convergence semiangle, acceleration voltage and aperture shape were all set according to the experimental geometry.

In contrast to previous applications of the FRFPMS method, we did not use molecular dynamics to generate snapshots of the vibrating structure. Instead, we first performed DFT simulations of the phonon eigenmodes (see below). Using this information, we have generated structure snapshots in an approach following refs. 60,61 by calculating atomic displacements owing to random excitation of phonon modes following their thermal population at 300 K. However, instead of summing over all of the phonon modes, we have split them by their eigenfrequencies into the above-mentioned 34 frequency bins and generated sets of 128 snapshots for each frequency bin separately. Considering the small unit cell of NiO, this approach brings DFT-level precision at a lower computational cost than training a machine-learning inter-atomic potential for subsequent molecular dynamics simulations.

**DFT calculations.** DFT calculations were performed using VASP[62] at the meta-GGA level using the r²SCAN functional[63] with the PAW pseudopotentials[64] containing the kinetic-energy density of core electrons. A 2 × 2 × 2 supercell of the conventional standard Fm$\bar{3}$m unit cell of NiO was geometrically optimized; the cell shape, cell volume and atomic positions were allowed to relax to a tolerance of 1 meV Å$^{-1}$, to capture the distortion away from the cubic symmetry caused by antiferromagnetic ordering along the [111] direction. For all calculations, a Γ-centred $k$-point grid with spacing 2π(0.06) Å$^{-1}$ was used with a plane-wave cut-off of 750 eV. The Python package phonopy[65,66] was used to generate the displacements required to calculate force constants in a 4 × 4 × 4 supercell of the conventional standard unit cell. The dielectric constant and Born effective charges were also calculated in a 4 × 4 × 4 supercell using the finite differences approach. These are for use in the non-analytical correction[67–69], required because of the polar nature of NiO.

Phonon modes were sampled on a grid of 5 × 5 × 49 **q** points spanning the Brillouin zone of a 2 × 2 × 2 supercell of NiO used in DFT simulations for calculation of the force matrix. The grid was chosen in a way to guarantee that atomic displacements are periodic across the boundaries of the simulation supercell used in phonon EELS simulations (see above).

## Data availability

The data that support the findings of this study are available from the corresponding authors on request.

## Code availability

All of the codes used in this work are available from the corresponding authors on request.

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

**Acknowledgements** SuperSTEM is the National Research Facility for Advanced Electron Microscopy supported in part by the Engineering and Physical Sciences Research Council (EPSRC) under grant number EP/W021080/1. We acknowledge further financial support from the EPSRC through grant numbers EP/V048767/1, EP/Z531194/1 and EP/V036432/1, as well as the Royal Society through grant no. IES/R1/211016. We acknowledge the Swedish Research Council (grant no. 2021-03848), Olle Engkvist Foundation (grant no. 214-0331), STINT (grant no. CH2019-8211), Knut and Alice Wallenberg Foundation (grant no. 2022.0079) and eSSENCE for financial support. The simulations were enabled by resources provided by the National Academic Infrastructure for Supercomputing in Sweden (NAISS), partially financed by the Swedish Research Council through grant agreement no. 2022-06725.

**Author contributions** D.K. and Q.M.R. designed and performed the experiments and analysed the experimental results. J.C.I. analysed the experimental results. J.Á.C.-R., P.M.Z., A.B. and J.R. designed and performed magnon calculations. A.K., J.A.d.N., J.R., P.M.Z., B.G.M. and V.K.L. designed and performed phonon calculations. K.E.h. prepared samples for analysis. All authors contributed equally to the analysis, interpretation and preparation of the manuscript.

**Competing interests** The authors declare no competing interests.

**Additional information**
**Correspondence and requests for materials** should be addressed to Demie Kepaptsoglou, Ján Rusz or Quentin M. Ramasse.

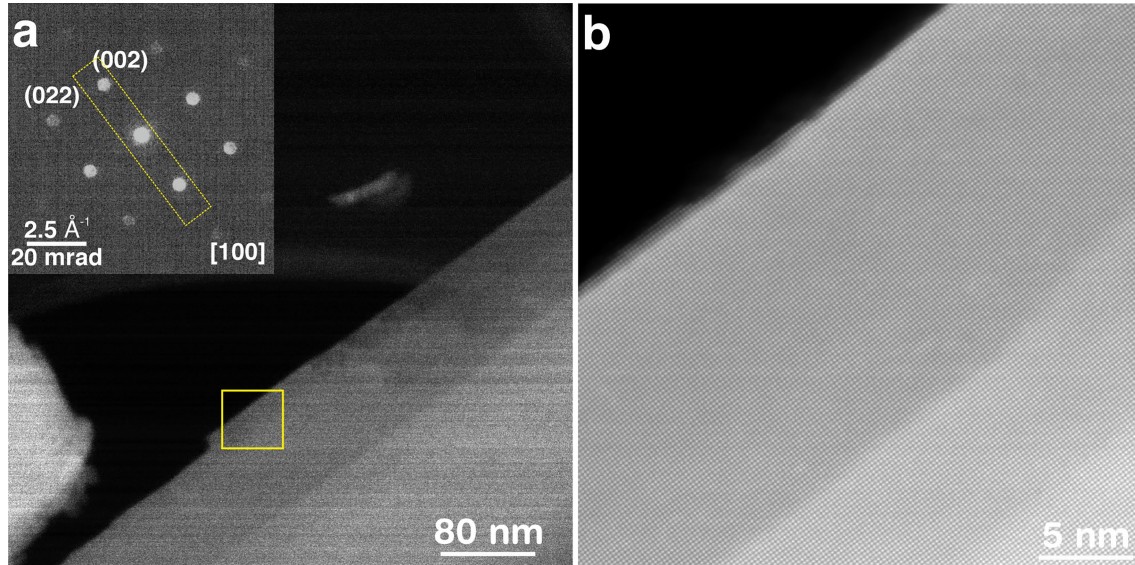

**Extended Data Fig. 1 | Imaging of the NiO single-crystal sample. a**, Annular dark-field image of the edge of a NiO single crystal, acquired with a 2.25 mrad semi-convergence angle (about 1.3 nm probe) along the [100] zone axis. Inset, experimental diffraction pattern along the NiO [100] zone axis at a 2.25 mrad semi-convergence angle, with the monochromating slit inserted, showing the orientation of the EELS collection slot aperture along the (002) row of reflections (dashed yellow box). **b**, Atomic-resolution high-angle annular dark-field STEM image acquired using a 31 mrad semi-convergence angle from the area marked with a yellow square in panel **a**.

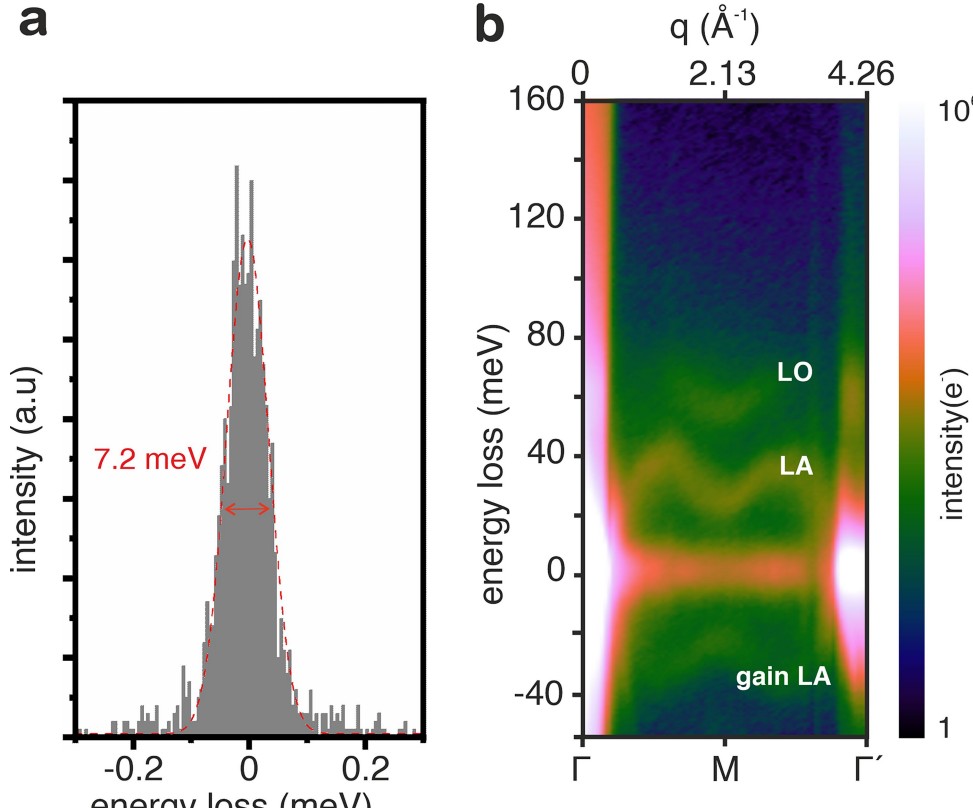

**Extended Data Fig. 2 | Vibrational EELS measurements of NiO. a**, EELS spectrum corresponding to a single acquisition frame (75 ms) in vacuum, showing a ZLP measuring 7.2 meV at the FWHM. **b**, As-acquired $\omega$–**q** maps along the 220 row of NiO reflections, showing the dispersion of the NiO LA/LO phonon branches, as well as the LA gain branch, presented on a logarithmic intensity scale. The dataset corresponds to 15,000 integrated frames (75 ms each).

**10 K**

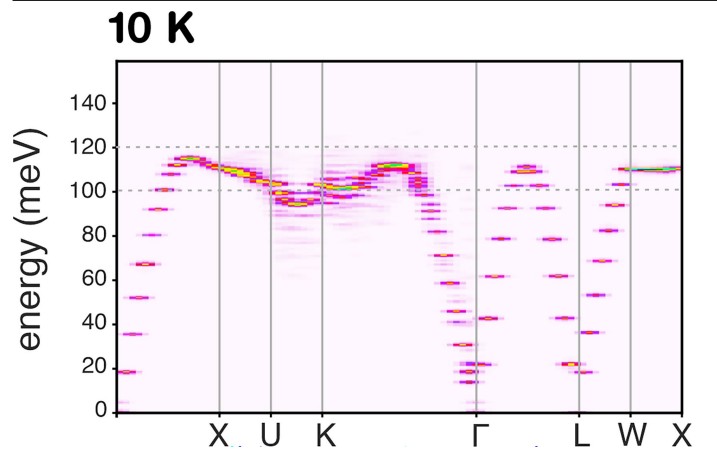

**300 K**

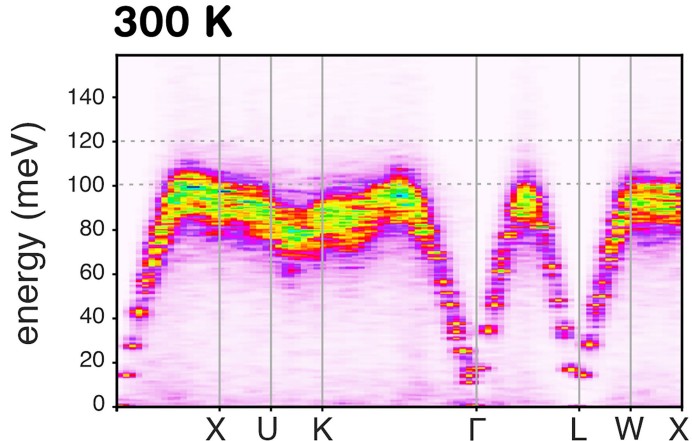

**Extended Data Fig. 3 | Magnon dispersions at different temperatures.**
Calculated magnon dispersions corresponding to the dynamical structure
factor computed using UppASD[34] for a NiO supercell consisting of $32 \times 32 \times 32$
repetitions of the cubic unit cell with periodic boundary conditions applied in
all directions. The ASD simulations were performed using a time step of 0.1 fs
over a total simulation time of 15 ps, with the remaining parameters the same as
in the main text.

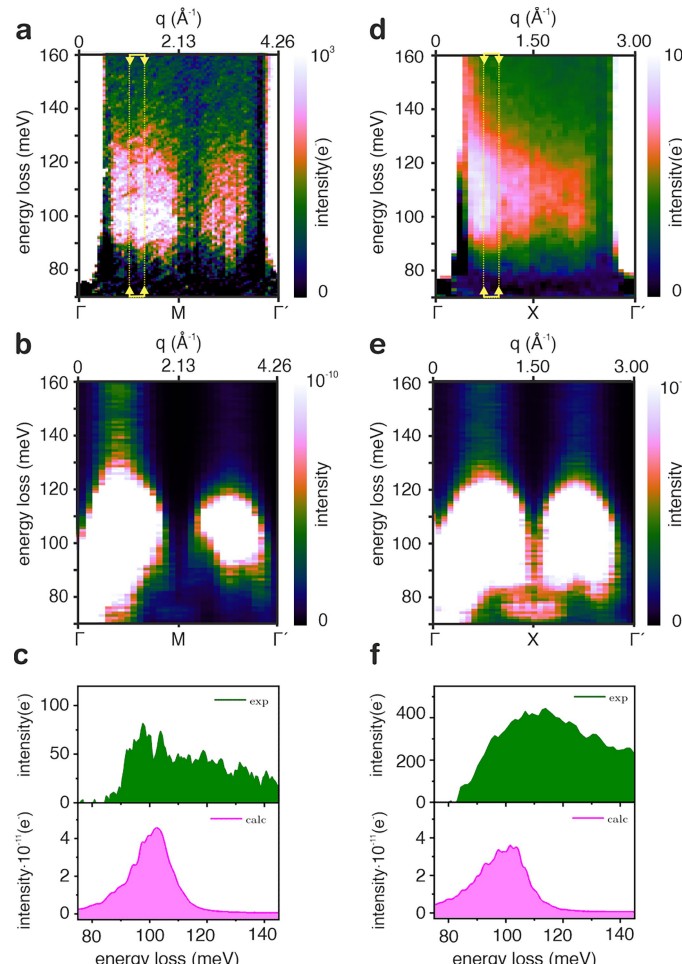

**Extended Data Fig. 4 | Experiment versus theory. a**,**b**,**d**,**e**, Experimental background-subtracted (**a**,**d**) and calculated (**b**,**e**) $\omega-\mathbf{q}$ EELS maps showing the dispersion of the magnon bands above 100 meV. **c**,**f**, Extracted spectra over a narrow momentum window ($\Delta q = 0.22\ \text{Å}^{-1}$), to avoid spectral broadening through momentum averaging, at the value at which the intensity is maximum along $\Gamma \rightarrow M$ ($q = 1.24\ \text{Å}^{-1}$) and $\Gamma \rightarrow X$ ($q = 0.97\ \text{Å}^{-1}$), marked by the yellow arrows and dashed lines in **a** and **d**, respectively. Calculated spectra at the same wave vector and averaged over a similar momentum window highlight the shape of the peaks, with a rising edge from 80 meV reaching a maximum at about 100 meV, before a weaker feature extending up to 120 meV, with the peak observed along $\Gamma \rightarrow X$ being broader and more rounded.