## [Peer Review File · Nature]

Magnon spectroscopy in the electron microscope

Corresponding Author: Professor Quentin Ramasse

Version 0:

Reviewer comments:

Referee #1

(Remarks to the Author)

Using high-resolution electron energy-loss spectroscopy (EELS), the authors present a study of magnon dispersion measurement on NiO, a typical antiferromagnetic material. They find a signal between 80-120 meV, which can be attributed to the magnon excitations according to the magnon diffusive scattering simulation underpinned by atomistic spin-dynamics simulations. The result demonstrates the methodological developments in STEM-EELS to expand the ability on spintronics researches. The experimental techniques are cutting-edge, and the results are well corresponded to the theoretical simulation. I have some comments to help the author further improve the manuscript:

1. Although the manuscript proposes several application scenarios for nanoscale detection of magnon, the experiments do not give truly spatially resolved data. How is the robustness of magnon? Will it change on the defects, surfaces, hetero-interfaces, or sample thickness/size? For such kinds of EELS experiments, one of the most important advantages is the high space resolution for the dispersion measurement, which is inaccessible to other conventional spectroscopies. For example, approaching the NiO-YSZ heterointerface, how does the magnon change? I recommend the authors include space-resolved data and related discussion to strengthen the technological breakthrough.
2. Although the authors demonstrate the measurement of magnon dispersion, it seems the extremely low detection efficiency of magnon limits further experiments and wide applications. Can authors give several possible outlooks on how to solve this problem in the future?
3. This is actually related to the above two comments. Considering the acquisition efficiency is low, and if we can not obtain space-dependent information, then what's the particular reason we should use this method for magnon dispersion measurement?
4. The relation between energy and momentum is probably the most important information of a dispersion, which is also a strong evidence for magnon excitation in the explanations of EELS spectra. However, such features are hard to recognize in the intensity map in Fig.2. More detailed analysis and line plots on the feature above 80 meV should be very helpful. For example, how do the peak position, linewidth and intensity change with momentum? I suggest adding these corresponding line plots.
5. In Fig.2 (b) and Fig.2 (e), the yellow lines that are guided to eye on dispersion measurement, representing the intensity of integrated signal. Usually, the guide to eye line represents the energy-momentum relation. This may cause confusion to the readers. It may be better to plot the intensity separately.
6. According to Fig.3 (b, d), the magnon intensity should be weaker at M/X point compared to the center of -M/ -X lines, which is not in consistence with yellow lines in the experimental data in Fig.2(b, e). The difference should be mentioned and explained.
7. I am curious is there any difference in magnon intensity between the 1st BZ and higher BZ? For example, is there any extinction phenomenon?
8. When comparing the previous inelastic neutron scattering result, the author explains the difference in energy is due to the temperature difference. The manuscript does mention that lower-temperature simulations give corresponding energies, but no specific results are shown. Could the author show how the magnon energy changes with temperature theoretically? This should enhance the result as magnon excitation.
9. In the present experiment result, the author used a small convergence angle to get the dispersion of magnon. However, a more common setting of STEM-EELS is to use a large convergence angle beam to gain high spatial resolution results. Will magnon signal be visible under large convergence angle beam? Is it related to on-axis or off-axis collecting geometry? This should provide valuable information to the EELS researchers.
10. The author claims the spatial resolution and momentum resolution under 2.25 mrad beam is 1.3 nm and 0.009\AA^{-1} . How did the author choose the convergence angle? How was the resolution evaluated?
11. Regarding background subtraction, the ω -q datasets are scaled by the square of the energy loss and integrated spectra

at the specific momentum positions are processed using a power-law background. Have you tried other background removal methods? Please evaluate and discuss the robustness.

Overall, I think this work is of great impact and will be of broad interest to the field of spectroscopy, spintronics and other condensed matter physics. The authors' contributions and findings are of considerable interest. Addressing the aforementioned questions would further enhance the clarity of this remarkable research.

Referee #2

(Remarks to the Author)

Summary of the Key Results

The submitted manuscript describes the measurement of magnon modes in a material suitable for such a first attempt: NiO. Dispersion curves of phonon and magnon modes are obtained using a new-generation monochromated scanning transmission electron microscope. The signal resulting from the scattering of a fast electron beam during interaction with these elementary excitations is analyzed with very high energy resolution and collected with a direct detection hybrid detector, maximizing the signal-to-noise ratio and enabling the experimental conditions necessary for detecting the magnonic signal, which has cross-sections several orders of magnitude lower than those of phonon modes. The dispersion curves are acquired under conditions approaching the fundamental diffraction limit, offering the best compromise between spatial resolution and wave vector q -resolution. Thus, this detection is achieved with nanometric spatial resolution, which is undeniably useful for future experiments aiming to probe magnetic inhomogeneities at these scales. The experimental results are corroborated by numerical simulations based on a well-suited theoretical approach.

Originality and Significance

The results obtained are a remarkable experimental achievement. They are convincing and significant for the perspectives they open in the field of magnonics. This is a groundbreaking experiment that validates recent theoretical predictions regarding the possibility of probing magnon modes with a fast electron beam. Following the initial demonstrations on phonons about a decade ago, which are now widely detected globally in electron microscopy down to atomic resolution, this work represents an important milestone in the quest to probe elementary excitations at "low energy" (typically in the THz frequency range) with high spatial resolution via inelastic scattering of fast electrons in an electron microscope. Here, a (purely) magnetic excitation is probed, which inherently exhibits very low intrinsic coupling with a fast electron beam, typically a privileged probe for excitations mediated by Coulomb interactions. This major result has been made possible by recent technological advancements in electron optics and detection sensitivity, positioning EELS as an advantageous alternative to "traditional" techniques for detecting magnetic excitations, such as inelastic neutron scattering. The perceived advantages include multiple orders of magnitude improvement in spatial resolution for probing magnetic responses and structures at the nanometer scale, requiring smaller sample quantities and significantly reduced acquisition times, and a much more versatile experimental setup (with possibilities for multimodal approaches within the same instrument: imaging, diffraction, elemental and chemical analysis, measurement of phonon modes at the nanometric or even atomic scale, etc.). The potential is even more promising for exploring related excitations (e.g., electromagnons or polarons), for which the expected cross-sections can be significantly higher.

Data and Methodology: Validity of Approach, Data Quality, Quality of Presentation

At this stage, this work remains a proof of concept on a material with favorable characteristics, including minimal spectral overlap between phonon and magnon bands. This is an evidently wise choice. The demonstration of the possibility of isolating the purely magnetic excitation signatures from any phononic or hybrid excitation remains to be achieved in the future.

The core of the demonstration relies on the acquisition and analysis of phononic and magnonic dispersion curves. Some corrections and additional information could improve the manuscript's presentation and clarity.

- Figure 2b: The superimposed yellow lines obscure the experimental data, preventing clear visualization. This should be improved by opting for a better graphical choice (e.g., a dotted and thinner line).
- Wave Vector Integration Range: The specific choice of the wave vector integration range for visualizing the partial magnonic DOS presented in Figures 2c and 2f does not seem to be justified in the text. Could the authors comment on this choice? How does these DOS features compare to a more broadly integrated DOS (while obviously avoiding regions near the Gamma points)?
- Simulation Approach: The authors use simulations based on a semiclassical approach derived from a standard multislice method, commonly applied when Coulomb interactions dominate. Although the technical details of this approach have been already published, it would be helpful here to describe, without delving into formalism, the physical (and not technical) elements included in the simulations that support the attribution of a magnonic signature to the measured experimental signal.
- Comparison of Experimental and Simulated DOS: It would be useful to compare the experimental and simulated partial DOS. Could the authors provide this additional information?

References : the authors provide a comprehensive and appropriate review of the use of particle beams to probe magnetic excitations, highlighting the benefits of EELS in terms of spatial resolution improvement. It would be beneficial to expand this review to include spatially-resolved techniques and mention the implementation of vector magnetometry, which is an essential tool in characterizing the distribution of currents and magnetization across a wide range of systems. Point defect sensors, like the nitrogen vacancy (NV) center in diamond for example, have shown impressive sensitivity and spatial resolution for detecting these fields, and are crucial for characterizing magnetic textures such as skyrmions or cycloids with varied morphologies, as well as imaging complex antiferromagnetic orders at the nanoscale.

Referee #3

(Remarks to the Author)

Detecting magnons using TEM-EELS is a valuable approach for investigating the spin dynamics and magnetic properties of materials. However, a key challenge lies in the relatively weak inelastic scattering cross-section of magnons compared to phonons, as both types of excitations typically occupy the same energy range in most materials. In this study, the authors demonstrate the detection of a magnon band at approximately 100 meV in an NiO sample, with the assignment corroborated by simulations. This is a challenging experiment with intriguing results, possibly give high energy magnon dispersions with spatial resolution, beyond the inelastic neutron scattering approach. However, the broader significance of this work needs to be better clarified.

Firstly, the separation of magnon and phonon signals in NiO is achievable due to their specific energy difference and the E^2 amplification factor. To strengthen the impact, it would be beneficial to discuss how this method might apply to other materials or heterostructures with less pronounced energy separations.

Many spintronic applications rely on phonon-magnon coupling in the low-energy regime, where phonon and magnon excitations mix. Therefore, examining cases where the phonon spectrum varies across magnetic phase transitions with high spatial resolution could add further relevance and interest.

Figure 2 and its caption lack clarity. The 220 lines appear nearly symmetric around the M point, while the 002 lines are asymmetric. A better explanation is needed for the asymmetry observed in panels b and e; is it an intrinsic effect of momentum-dependent scattering or a result of experimental limitations? Additionally, if the green areas in panels c and f represent integrated contributions, it is unclear why they drop to zero below 70 meV, unlike panels a and d. The reliability of subtracting the power-law background from the LO phonon signal to extract the weak magnon signal should be addressed, as should the potential for shifting the arrowed regions leftward to emphasize stronger signals.

The agreement between theory and experiment could be improved, as discrepancies are noticeable: the spectra in Fig. 3b and 3d differ from those in Fig. 2b and 2e. It would be helpful to present the magnon bands, equations of electron-magnon scattering cross-sections, and perhaps an animation of the 100 meV magnon mode to better illustrate theoretical advancements and support reader comprehension.

Version 1:

Reviewer comments:

Referee #1

(Remarks to the Author)

I would like to thank the authors for their nice work in addressing the review comments, particularly regarding my primary concerns about the robustness and reliability of the "signal". The addition of the new Figure 4, which demonstrates the spatial distribution of magnon signals, significantly strengthens the paper. The new data clearly show that the "signal" is exclusively present in the NiO regions, providing spatial evidence to support the identification of these signals as magnon-related. Other improvements to the manuscript have also enhanced its overall quality.

However, some of my questions about how surfaces and interfaces influence magnon behavior remain unanswered. This limitation may stem from the current signal-to-noise ratio constraints caused by the small scattering cross-section of magnons in the EELS experiments. While these scientific questions could be explored in future studies, they do not diminish the value of this work, which focuses on methodological demonstration and conceptual validation. Overall, the robust demonstration of magnon dispersion measurement represents a significant breakthrough in the fields of physics and electron microscopy.

My remaining concern lies in strengthening the evidence for distinguishing magnon signals from other possibilities, especially given the notable discrepancies between simulations (Fig. 3b,d) and experimental data (Fig. 2c,d). While the intensity variations in Fig. 2c show partial agreement (e.g., low intensity at the M-point), the strong intensity at the X-point in Fig. 2d contradicts Fig. 3d, which the authors attribute to insufficient momentum resolution. The energy discrepancies between simulations and experiments (peaks between Γ -M/X in simulations versus no clear change along Γ -M- Γ and monotonic change along Γ -X- Γ in experiments) also require clarification. I wonder whether these phenomena are reproducible and reliable.

To further enhance reliability, the authors may consider the following suggestions:

1. In fact, I was expecting that authors have performed the spatial mapping similar to Figure 4 but with a smaller convergence angle (using a slit aperture rather than the round one), which could reveal how the signals distribute in space across different momentum points (e.g., Γ /M/X-points). Current limitations in momentum resolution (due to the large convergence semi-angle ~ 31 mrad in dark-field EELS) prevent such analysis. For example, if the "signal" is magnon-related, stronger intensity should appear in NiO regions when the momentum is between Γ and M, while weaker signals would occur at Γ and M points based on experimental data Fig.2c and simulated data Fig.3d. These momentum-resolved spatial

mapping could also clarify whether discrepancies arise from experimental or simulation limitations.

2. In Extended Data Figure 4, presenting spectra from additional momentum windows (e.g., X/M-points) alongside existing data would provide intuitive evidence of magnon dispersion. Direct comparisons between these spectra could strengthen the interpretation.

3. Given the overlapping energy ranges between magnons and amorphous carbon vibrations, authors may need to discuss and exclude this possibility. Moreover, discussing reproducibility is also needed. In fact, through repeated experiments and stacking multiple datasets would improve confidence in the validity of magnon dispersion. With such improvements, the energy variations in the dispersion curves may show better consistence between experiments and simulations.

Referee #2

(Remarks to the Author)

The version resubmitted by the authors is significantly improved both in terms of presentation and scientific content. The work presented is enhanced by a new high-spatial resolution mapping experiment of magnon modes in a thin NiO layer. The manuscript now combines two remarkable types of experiments: spatially-resolved and momentum-resolved measurements. The responses and modifications made to my questions and suggestions are particularly relevant.

In conclusion, the work presented is of very high quality and great importance; it represents a significant milestone in the field of magnonics.

Referee #3

(Remarks to the Author)

The author provided comprehensive responses to my previous questions, making the paper more suitable for consideration by Nature. However, my main concern remains: the TEM-EELS technique, with its poor energy resolution, does not offer a clear advantage for probing magnons. While its spatial resolution is arguably its only advantage over neutron probes, it does not provide insight into local variations in exchange interactions or magnetic anisotropy at surface or interface. The absence of magnons in YSZ is unsurprising. The author should better demonstrate the broader usefulness of these measurements beyond being a first-of-its-kind study.

Additionally, the meaning of the green regions in Figures 2e and 2f should be clarified in the caption. Surprisingly, these panels were even not discussed in the text.

Version 2:

Reviewer comments:

Referee #1

(Remarks to the Author)

Upon careful review of the authors' response to the reviewer comments, the newly added experimental data (e.g., Supplementary Fig. 4 and Supplementary Fig. 5) and expanded explanations have basically addressed my concerns. While the anticipated spatial mapping data using a small convergence angle and slit aperture were not obtained, this limitation is understandable given the signal-to-noise ratio (SNR) constraints: for instance, the moderate SNR in Fig. 2 would degrade significantly when distributing signals across spatial pixels, making meaningful information extraction unfeasible. So, it seems that currently meaningful data can only be obtained by using a large convergence angle.

That said, I remain confident that using the small convergence semiangle and slit aperture with improved SNR through technical advancements in future, could fully resolve the previously mentioned signal self-consistency issues and enable studies of surface/interface effects.

Overall, I recommend the paper for publication, as it represents a breakthrough in electron microscopy—expanding its capabilities to investigate magnetic phenomena beyond established techniques such as Lorentz microscopy, DPC, ptychography, holography, EMCD, and core-loss spectroscopy and so on. However, I do suggest the authors include a discussion or outlook on unresolved questions in the manuscript to highlight the research's open challenges, some of which I have mentioned above.

Regarding the editor's note on statistical analysis and error bars, the current data lack such details. Given the authors' mention of Figure 2a being the sum of 90,000 frames (and similar data processing for others), incorporating statistical error analysis (e.g., standard deviation, consistency across repeated measurements) for key results would strengthen the reliability of their conclusions.

Response to Reviewers – Manuscript 2024-10-21687

Referee #1 (Remarks to the Author):

Using high-resolution electron energy-loss spectroscopy (EELS), the authors present a study of magnon dispersion measurement on NiO, a typical antiferromagnetic material. They find a signal between 80-120 meV, which can be attributed to the magnon excitations according to the magnon diffusive scattering simulation underpinned by atomistic spin-dynamics simulations. The result demonstrates the methodological developments in STEM-EELS to expand the ability on spintronics researches. The experimental techniques are cutting-edge, and the results are well corresponded to the theoretical simulation.

We are delighted to read such high praise from the reviewer and we thank them for their thoughtful and constructive comments, which we hope to have addressed in the responses below.

1. Although the manuscript proposes several application scenarios for nanoscale detection of magnon, the experiments do not give truly spatially resolved data. How is the robustness of magnon? Will it change on the defects, surfaces, hetero-interfaces, or sample thickness/size? For such kinds of EELS experiments, one of the most important advantages is the high space resolution for the dispersion measurement, which is inaccessible to other conventional spectroscopies. For example, approaching the NiO-YSZ heterointerface, how does the magnon change? I recommend the authors include space-resolved data and related discussion to strengthen the technological breakthrough.

*We would like to thank the referee for their suggestion to further explore spatially resolved magnon signals as a way to strengthen the impact of this manuscript. Spurred by this comment we have now included an additional figure (Figure 4), displaying a newly acquired spectral map on the NiO/YSZ system. The two-dimensional map demonstrates the **spatially localised** presence of the magnon feature at ~100meV (panel d in Figure 4), confined solely within the NiO film, with the signal disappearing completely in the (non-magnetic) YSZ substrate and in vacuum.*

The new measurements were obtained using the dark field EELS geometry [reference 20 in the original submission, F.S. Hage, et al., Physical Review Letters 122, 016103 (2019)], i.e. with a highly convergent, angstrom-sized electron beam. This geometry was predicted in our theoretical parameter exploration to be favourable for extracting spatially resolved magnon signals [reference 28 in the original submission, K. Lyon et al., Physical Review B 104, 214418 (2021)], as was the case for phonon spectroscopy (although the magnon scattering physics are different from the phonon case, of course).

This is the first demonstration of nm scale mapping of magnons in an electron microscope: the NiO film is only 30nm wide, and the observed signal attributed to magnon scattering disappears within a sub-nm distance of the interface or the vacuum edge.

Furthermore, simulations for a thin slab of NiO predict a smaller intensity of the magnon signal at its edges compared to the centre of the slab, with a broader, less 'spiky' post-peak tail, something with which the experimental data seems not to be in disagreement. This is further illustrated in a new supplementary figure included in an online-only supplementary document. However, we should emphasise that these observations could have other causes (e.g. signal broadening, or simple drift), and do not represent a conclusive fingerprint of a spatially resolved fine structure change of the magnon peak.

We agree that further investigating how magnons evolve near heterointerfaces and at (atomic-scale) defects is of great interest to the community and represents a compelling direction for future research. However, due to the signal strength limitations (see the discussion in response to question 3), we are not yet in a position to report conclusive spectroscopic fine structure differences close to the interface or at defects at the atomic scale. We also note that the interpretation of such signal differences would require the additional development of appropriate modelling frameworks, and that we are far from full theoretical descriptions.

In addition to these new data, we would like to clarify that the nanoscale momentum-resolved results presented in the original submission were acquired over regions of size $2 \times 2 \text{ nm}^2$. This resolution enables the study of magnon dispersions in nanometre-scale samples, demonstrating the feasibility of examining localized phenomena in materials with a spatial precision that is challenging to achieve with other conventional spectroscopies.

Together with the addition of the spatially resolved spectroscopy data, we hope that this will help to further convince the reviewer and prospective readers that this is an exciting development in the fields of electron microscopy and magnonics. The initial demonstration of magnon detection and dispersion at this resolution will enable future studies aiming to explore spatial variations in magnon behaviour near heterointerfaces and other nanoscale features in greater detail.

Author actions:

- A new Fig. 4 describing spatially resolved magnon dark-field EELS measurements across a NiO thin film has been introduced.
- The associated section in the main text (lines 192-218) and discussion (lines 231-235) sections of the manuscript have been expanded to reflect the new figure addition and supporting theoretical calculations.
- New Supplementary note 3 and corresponding Supplementary Fig. 3 outline theoretical calculations that explore the spatial dependence of magnon scattering across a NiO slab.

2. Although the authors demonstrate the measurement of magnon dispersion, it seems the extremely low detection efficiency of magnon limits further experiments and wide applications. Can authors give several possible outlooks on how to solve this problem in the future?

As the referee correctly remarks, the weak nature of the magnon signal remains a limitation, especially in systems where phonon and magnon signatures overlap.

We do provide in the original submission (lines 229 to 254) a discussion of possible mitigation strategies and we sketched an outlook for further and wider deployments of this technique in the future, in line with its intrinsic limitations. We apologise if these remarks were not clear enough: we have now amended the wording to highlight and to clarify the intent of this section of the discussion.

To summarise, we think that four main factors, which have been highlighted in the revised manuscript, are likely to contribute to mitigating the low detection issues.

- First and foremost, **increasing detection time** will improve signal levels. This approach is widely employed across various advanced spectroscopy techniques, including at X-ray synchrotron sources, neutron spallation facilities, and even for astronomical observations with the Hubble and James Webb Space telescopes. In these cases, extended acquisition times, sometimes to

extremes (days or even weeks of acquisitions for single datapoints) allow for the detection of faint or subtle signals with enhanced signal-to-noise ratios.

Electron energy loss spectroscopy usually involves far shorter acquisition timescales – mostly to retain the technique’s spatial resolution advantage. As we explain in our manuscript, continuous hardware improvements (sample stages, monochromators, higher efficiency direct electron detectors) will allow longer acquisition times, and therefore higher fidelity of magnon detection at high spatial resolution in the (near) future.

- *Secondly, the use of **additional experimental parameters** can enhance or suppress the intensity of the magnon signal relative to vibrational (phonons) or other spectral contributions. Variable temperatures (down to liquid He temperature and up to elevated temperatures), or the use of magnonic device architectures in the electron microscope (e.g. through electrical biasing via MEMS-technology sample holders) can be leveraged to purportedly generate high spin currents and enhance the magnon signal, leading to higher detection efficiency.*
- *Several of the materials of interest for magnonics, including **metallic and oxide systems, present favourable separation between signals and higher intrinsic magnon scattering cross-sections**. There are therefore plenty of additional systems to explore as the technique further develops, where low detection efficiency is not prohibitive.*
- *Finally, we also mention in our original submission the potential of exploiting the **interaction of magnons with other quasi-particles** whose scattering involves energy losses in the same range. These interactions can lead to enhanced signals or unique dispersive behaviours due to quantum effects (anti-crossing behaviour of their respective dispersive branches), which could be directly exploited to fingerprint magnon scattering with signal strength presenting less of a limitation.*

Author actions:

- The Discussion section (lines 244-272) has been expanded and reworded to highlight strategies to circumvent limitations due to low magnon signal, and to clarify the intent of this section of the discussion.

3. This is actually related to the above two comments. Considering the acquisition efficiency is low, and if we cannot obtain space-dependent information, then what’s the particular reason we should use this method for magnon dispersion measurement?

We thank the reviewer for this thoughtful comment about the rationale for using this method to detect magnons, given current practical challenges. We hope that our answers to the two previous questions address this point, having demonstrated how spatially resolved information is in fact already obtainable, while outlining a clear roadmap for improvements in detection strategy to maximise signal detection.

To reiterate, we believe there is no fundamental physical limitation to prolonging acquisition times (as long as beam voltage and current are maintained below damage threshold limits for a given material), which will allow us to overcome challenges related to detection efficiency. By extending acquisition times, we can achieve the necessary signal-to-noise ratio for high-quality magnonic data without compromising spatial resolution.

*Additionally, we would also like to emphasize that a key advantage of using EELS in STEM is its unique ability to probe magnonic signals **at the nanoscale**, including in individual nanoparticles, even in cases*

where actual spatial resolution is, comparatively speaking, limited. This capability is in stark contrast to, e.g., neutron spectroscopy, which, while highly effective for studying magnons in bulk materials, requires single crystals with large sample volumes (orders of magnitude larger than for STEM, sometimes difficult to synthesise or not relevant in the context of nanoscale device physics) and is inherently unsuitable for nanoscale investigations. A compelling parallel is the widespread application of STEM in studying the electronic structure of 2D materials, where neutrons are similarly limited due to sample size constraints.

This combination of high spatial resolution and the flexibility to optimize acquisition strategies for improved detection efficiency also makes EELS in STEM particularly well-suited for magnon dispersion measurements, especially in contexts where sample size, heterogeneity, or spatial selectivity is of importance. Although momentum resolution is intrinsically limited in this nanoprobe regime [a full discussion can be found in reference 25 in the original manuscript, F.S. Hage et al., *Science Advances* 4, eaar7495 (2018)], the combined (balanced) momentum and spatial resolutions can play a crucial role in understanding the underlying scattering physics in nano-scale objects. Indeed, acoustic magnon modes are highly dispersive at the Gamma point and their study thus requires momentum selectivity. We would therefore argue that both types of measurements—spatially and momentum-resolved—are essential for a comprehensive understanding of the spin wave propagation, as well as other effects such as phonon magnon coupling. This combination is only possible in the STEM.

Author actions (see also response to point above):

- A new Fig. 4 describing spatially resolved magnon dark-field EELS measurements across a NiO thin film has been introduced.
- The associated section in the Main text (lines 192-218) and Discussion (lines 226-235) sections of the manuscript have been expanded to reflect the new figure addition and supporting theoretical calculations.
- New Supplementary note 3 and corresponding Supplementary Fig. 3 outline the theoretical calculations of the spatial dependence of magnon scattering across a NiO slab.

4. The relation between energy and momentum is probably the most important information of a dispersion, which is also a strong evidence for magnon excitation in the explanations of EELS spectra. However, such features are hard to recognize in the intensity map in Fig.2. More detailed analysis and line plots on the feature above 80 meV should be very helpful. For example, how do the peak position, linewidth and intensity change with momentum? I suggest adding these corresponding line plots.

Thank you for this suggestion! We agree wholeheartedly. In an attempt to combine information in a single figure (figure 2) in the original submission, this essential information and associated analysis became difficult to distinguish.

*The variation of the magnon signal in the experimental data along the momentum axis is now illustrated more clearly in new and updated panels in Figure 2, now labelled Figure 2c and 2d. These display the energy range in the ω - q map corresponding to magnon losses only, where in order to better separate the magnon signal, the background from the decaying tail of the phonon peaks at lower energy losses was removed. A first order log polynomial was used in this instance, a background model often used in the low loss region due to its slower decay tail (see, e.g., R. Egerton, *Electron Energy Loss Spectroscopy in the Electron Microscope*, Springer, 2011). More comments on background subtraction will be provided in response to question 11 below.*

The original version of Figure 2, comprising the ω - q maps normalised by the square of the energy loss (so called “ E^2 plots”) has been moved to a new supplementary information file, together with a Supplementary note explaining the rationale behind the scaling by the square of the energy loss.

The correspondence with simulated data is now far more striking. Taking the example of the 220 data (see the updated Figs. 2c and 3b, or Extended Data Fig. 4), two magnon dispersion lobes of spectral intensity are clearly visible on either side of the M point, with a maximum peak at around 100meV. The lobes’ intensity is asymmetric on either side of the M point, with stronger signal in the first BZ (between Γ and M) compared to the lobe between M and Γ' . The intensity tends to 0 towards each of the BZ vertices (Γ , M and Γ'). While the asymmetry is also evident in the 002 data (see the updated Figs. 2d and 3d, or Extended Data Fig. 4), the separation between ‘lobes’ is less pronounced. This is not surprising as the Γ -X- Γ' distances are shorter in this direction, with the moderate momentum resolution and relatively low signal to noise ratio making it more challenging to obtain a clear separation between branches.

Limitations in signal strength preclude at this stage any meaningful analysis of the linewidth of these magnon branches. We note, however, that the same still holds in vibrational spectroscopy. Although remarkable progress has been achieved in vibrational STEM-EELS in the 10 years since its demonstration, to our knowledge, there has not yet been any report of quantitative linewidth analysis of the phonon branches (which would lead to a direct quantification of phonon lifetimes) in momentum-resolved experiments. We fully expect that, as these techniques mature and limitations with signal quality are overcome, such tantalising and exciting analyses will become readily possible.

Author actions:

- Figure 2 has been replaced with an updated version, with fewer overlaid plots, to improve readability. Panels c,d now include background-subtracted ω - q maps.
- The associated main section of the manuscript (lines 102-133) has been updated to reflect figure changes, including an expanded discussion of the magnon dispersion behaviour and signal asymmetry across the Brillouin zone.
- The original figure 2 is now included as Supplementary Fig. 1 to illustrate the variation of the magnon signal along the momentum axis through the E^2 scaling of the ω - q map, alongside a Supplementary note motivating this data scaling approach.
- A new Extended Data Fig. 4 has been introduced to facilitate comparison with theoretical calculations of magnon EELS.

5. In Fig.2 (b) and Fig.2 (e), the yellow lines that are guided to eye on dispersion measurement, representing the intensity of integrated signal. Usually, the guide to eye line represents the energy-momentum relation. This may cause confusion to the readers. It may be better to plot the intensity separately.

We agree with the referee that these lines were negatively impacting the figure readability. They have now been removed in the revised version of this Figure, which has been moved to a Supplementary material document for completeness, and replaced by new panels.

Their initial intended purpose was to illustrate the variation of the magnon signal along the momentum axis: this has now been addressed with additional figures and discussions, as explained in response to question 4 above.

Author actions (see also response to question 4 above)

- Figure 2 has been replaced with an updated version, with fewer overlaid plots, to improve readability. Panels c,d now include background-subtracted ω -q maps instead.
- The associated Main section of the manuscript (lines 102-133) has been expanded to reflect figure changes.
- The original figure, without the overlaid plots, is now included as Supplementary Fig. 1 to illustrate the variation of the magnon signal along the momentum axis through the E^2 scaling of the ω -q map alongside a Supplementary note motivating this data scaling approach.

6. According to Fig.3 (b, d), the magnon intensity should be weaker at M/X point compared to the center of Γ -M/ Γ -X lines, which is not in consistence with yellow lines in the experimental data in Fig.2(b, e). The difference should be mentioned and explained.

The referee is correct: the magnon intensity is indeed weaker (trending to 0) at the M and X points, as clearly shown by our simulations in Figure 3. This is also the case in experimental data, as detailed in response to questions 4 and 5, and we believe there is no inconsistency.

To reiterate, the poor readability of the original version of Figure 2 made this difficult to assess, for which we apologise. Having updated Figure 2, we believe it is now clear that the experimental signal is weak at the M and X points. Furthermore, the signal clearly peaks between Γ and X (resp. Γ and M), comparatively closer to X than to M, an observation also borne out by simulations.

We have also included a new Extended Data Figure 4, where experimental and calculated magnon dispersions are plotted side by side in the same energy range which we hope illustrates more clearly the agreement between experiment and theory.

Author actions (see also response to question 4 above)

- Figure 2 has been replaced with an updated version, now including background subtracted ω -q maps which illustrate more clearly the weaker intensity at the M/X points.
- The overlaid yellow line profiles, which were hindering legibility, have been removed.
- The associated section of the manuscript (lines 113-133) now describes in more detail the magnon signal variation in energy and momentum, with an excellent match to theoretical calculations.
- An additional extended data Figure 4 displays experiment and theory ω -q maps and spectra side by side, demonstrating the match between theory and experiment.

7. I am curious is there any difference in magnon intensity between the 1st BZ and higher BZ? For example, is there any extinction phenomenon?

Many thanks for this insightful question. Differences in intensity between the first and higher order BZ are indeed expected: this already is illustrated to an extent by the intensity asymmetry of the magnon 'lobes' on either side of the X and M points in both experimental and simulated EELS data, with the intensity being stronger in the first BZ.

Here, due to geometrical constrains in the experiment and to maintain energy resolution across the entire momentum selecting slit, our experiments do not extend beyond the second BZ (first order Bragg reflection).

However, the inelastic neutron scattering data and associated simulations in Reference 37 in the original submission [Sun, Q. et al. PNAS 119, e2120553119 (2022)] explore a wider momentum transfer

range. They point to a similar behaviour, with what seems to be a strong dampening of the magnon signal in the higher order BZs, although the specificity of the underlying physics for neutron scattering make the comparison of signal strength with EELS challenging (e.g., for inelastic scattering, due to scattering form factors the magnon intensity may be strongest between the first and second BZ before tailing off).

A full theoretical analysis would be necessary to term this an “extinction” – as this has a very specific meaning. Further complexity is also added in the case of EELS, due to plural scattering across momentum space affecting relative intensities, and due to the specific scattering physics with the probe electrons.

This will prove a fascinating avenue of research for the magnon-EELS field as it grows and develops.

Author actions:

- The relevant section of the manuscript (lines 113-118) has been to describe the intensity variation across the Brillouin zone.
- Lines 139-142 also link these observations to similar asymmetry observed in inelastic neutron scattering experiments.

8. When comparing the previous inelastic neutron scattering result, the author explains the difference in energy is due to the temperature difference. The manuscript does mention that lower-temperature simulations give corresponding energies, but no specific results are shown. Could the author show how the magnon energy changes with temperature theoretically? This should enhance the result as magnon excitation.

Many thanks for suggesting the inclusion of this additional analysis and simulation work. We agree it will be a useful reference for the community.

Previous theoretical works [Ref [41]; Woo, et al. Physical Review B 91, 104306 (2015)] show that the magnon density of states red shifts with the increase of temperature. Our own calculated dispersion curves for NiO are now included as Extended Data Figure 3. They confirm the red shift, together with a significant broadening of the magnon band when the temperature is increased to 300 K.

We note that the consequence of this result is that the use of cryogenic temperatures for magnon observations in STEM-EELS may prove favourable in future experiments (at least in the case of NiO), due to the increased energy separation between magnon and phonon signals in this system, as well as the expected suppression the phonon modes. A potential caveat of the use of cryogenic temperatures is of course temperature and stage stabilities, which could restrict acquisition times. Moreover, any realistic magnonic material system would need to operate at RT or above, so understanding properties at such conditions is necessary for future applications.

Author action:

- An Extended Data Fig. 3 comprising magnon dispersions at different temperatures has been introduced to highlight the agreement between experimental and calculated magnon EELS dispersion curves.
- The Main text section of the manuscript (lines 138-139 and 176-177) has been amended accordingly to reflect the introduction of the new figure.

9. In the present experiment result, the author used a small convergence angle to get the dispersion of magnon. However, a more common setting of STEM-EELS is to use a large convergence angle beam to gain high spatial resolution results. Will magnon signal be visible under large convergence angle beam? Is it related to on-axis or off-axis collecting geometry? This should provide valuable information to the EELS researchers.

We agree with the referee that the strength of STEM-EELS lies in its high spatial resolution, typically achieved using large convergence angles. We hope that our response to question 1 above answers this additional point satisfactorily.

The magnon signal is visible at higher convergence angles, which the new Figure 4 now illustrates clearly. Furthermore, our earlier theoretical exploration of optical parameters for magnon EELS [reference 28 in the original submission, K. Lyon et al., Physical Review B 104, 214418 (2021)] had suggested that a dark-field (off-axis) EELS collection geometry would be favourable for the acquisition of spatially resolved signal. This is the experimental configuration we chose for this additional experiment.

We note that the results in Lyon et al. also point to prospects in extracting atomic-level information – a direction of future research that we will explore as a next step.

Author actions:

- A new Fig. 4 describing spatially resolved magnon dark-field EELS measurements across a NiO thin film has been introduced.
- The associated section in the Main text (lines 192-218) and Discussion (lines 230-235) sections of the manuscript have been expanded to reflect the new figure addition and supporting theoretical calculations.
- New Supplementary note 3 and corresponding Supplementary Fig. 3 outline the theoretical calculations of the spatial dependence of magnon scattering across a NiO slab.

10. The author claims the spatial resolution and momentum resolution under 2.25 mrad beam is 1.3 nm and 0.009\AA^{-1} . How did the author choose the convergence angle? How was the resolution evaluated?

The chosen convergence angle of 2.25mrad in our electron microscope used provides a good balance of spatial (1.3nm diffraction limited probe) and momentum resolutions. Much larger beam sizes with near-parallel beams are an alternative strategy for momentum-resolved EELS, more akin to a pure diffractometer-based experiment, achieving very high momentum resolution, but without the spatial selectivity of our approach, which was intended to illustrate the advantages of using STEM-EELS for this kind of measurements.

While smaller nanoprobe convergence angles are readily achievable (we have also acquired data at 1.2mrad convergence in the process of exploring magnon spectroscopy), in practice and for our instrument, these optical settings place more strain on the electron optical set-up, making stability harder to achieve. As a result, a target of $\sim 2\text{mrad}$ represented an excellent compromise for nano-scale momentum-resolved experiments.

The spatial resolution quoted corresponds, for simplicity, to the size of a diffraction-limited beam with that 2.25mrad convergence and with a 60keV primary energy.

The momentum resolution calculation is described in the Methods section, and stems from the combined effect of the convergence and collection apertures used. This is often described using an 'effective' collection angle (the quadratic sum of the collection half angle β and convergence half-angle α). See, e.g., F.S. Hage et al., *Phys. Rev. B* 88, 155408, 2013.

To be exact, the momentum resolution can be expressed as $k\sqrt{\sin^2 \alpha + \sin^2 \beta}$, where k is the electron beam wavenumber, and α and β are the convergence and collection semi-angles, respectively.

In the case of a rectangular momentum-selecting EELS slit, which is here chosen to have a width in the energy-dispersive direction that matches the beam convergence, it is easy to see that the momentum resolution can for simplicity be considered to be limited by the size of the diffraction spots on the EELS camera (that is, the full beam convergence angle).

The value we provided in the original submission omitted a factor of 2π in the definition of distances in reciprocal space, as well as the electron wavenumber. While this convention is often used in crystallography, and all the numbers used in our submission were correct in this paradigm, after discussions with colleagues and to be consistent with numerous publications in momentum-resolved electron energy loss spectroscopy (including our own previous work), we have corrected all reciprocal space distances in our revised figures and the updated discussion to now include this factor or 2π . This also affects the calculation of momentum resolution. We estimate the effective momentum resolution with this convention as 0.4\AA^{-1} , consistent with our previous work reported with similar optical parameters in F.S. Hage et al., *Science Advances* 4, eaar7495 (2018).

Author actions:

- The rationale for the choice of convergence angle is now outlined in the Methods section, lines 437-441.
- The Methods section (lines 447-460) has been expanded to include full details of the determination of the momentum resolution.
- All dimensions along the reciprocal direction \mathbf{q} , momentum resolution in the Main text section (e.g. lines 105-137) and Figures have been updated to include a factor of 2π (previously omitted due to the use of a different convention for defining distances in reciprocal space).

11. Regarding background subtraction, the ω - q datasets are scaled by the square of the energy loss and integrated spectra at the specific momentum positions are processed using a power-law background. Have you tried other background removal methods? Please evaluate and discuss the robustness.

This is a very interesting question, which presents an on-going challenge for the entire high-resolution EELS community. The use of a power-law function to remove any decaying background is one of the most commonly used approach in EELS, as it models population statistics quite effectively and appears to be relatively robust, even in the vibrational EELS range [e.g., K.L.Y. Fung et al., *Ultramicroscopy* 217, 113052 (2020) or recently Haas B, et al., *Nano Lett.* 2023, 23, 13, 5975–5980]. However, the stability of the model across neighbouring pixels in 2-dimensional datasets can be poor when the energy fitting window is narrow (for very low energy losses close to the zero-loss peak tail, or for signals superimposed on the tail of other losses close in energy). This can lead to subjective results depending on a given user's fitting window selection. As a result, most research groups in the field employ

background subtraction on a case-by-case basis, considering various background options, which can sacrifice physical justification to improve visibility [Hachtel JA, et al., Sci Rep. 2018, 8 (1), 5637].

Here, we initially chose to present our magnon data with minimal user intervention. The scaling by the square of the energy loss is entirely user-agnostic, and while it has some physical justification (especially at very low energy losses: see a newly added Supplementary note), it can be thought of as a pure ‘data scaling’ strategy (as would a logarithmic intensity display be), helping to visualise on the same panel signals with vastly different intensity levels.

We had also used power-law subtracted spectra in a number of figures, solely as an illustration of the subtle magnon feature superimposed on the tail of the LO phonon.

However, as the consensus among reviewers indicated that the presentation of the magnon dispersion curves could be improved, we explored different fitting options for the removal of the LO phonon decaying background immediately before the energy range where our simulations predict the presence of the magnon signal. A new Supplementary Figure 2 presents the ω - q maps along the 220 systematic rows of reflections, using first-order logarithmic polynomial and power-law models for comparable energy windows.

In both cases, the resulting signal shows a remarkable qualitative resemblance to the calculated curves in Figure 3 of the main manuscript. However, the use of the power-law background appears to yield noisier maps, as it is more susceptible to variations in phonon intensity tail across the narrow fitting energy window, resulting in some negative values, particularly on the weaker M - F' magnon branch. Therefore, we have decided to retain the first-order logarithmic polynomial subtracted data for a new, update figure 2 in the main manuscript, and have updated the spectra in Figures 2e and 2f accordingly.

For completeness, the full ω - q maps, displaying the entire energy range and including the zero-loss line, phonon and magnon branches, both without scaling and with scaling with the square of the energy loss, as initially shown in our original figure 2, are also still presented for completeness as supplementary figure 1.

Author action:

- Figure 2 has been updated to include background subtracted ω - q maps (c,d) and spectra (e,f) produced using a log-polynomial background fitting method to model the decaying phonon intensity.
- Supplementary note 2 and corresponding Supplementary Figure 2 have been introduced to discuss the robustness of background subtraction as a function of the choice of model.
- Supplementary note 1 provides a discussion of the physical justification for E^2 scaling of data, alongside the original figure 2, for completeness.

Overall, I think this work is of great impact and will be of broad interest to the field of spectroscopy, spintronics and other condensed matter physics. The authors’ contributions and findings are of considerable interest. Addressing the aforementioned questions would further enhance the clarity of this remarkable research.

Once more, we would like to thank the referee for such kind words and for finding our research “remarkable” and of “considerable interest”. The very constructive comments and questions provided in the review have helped us clarify our manuscript, and we believe the presentation of the results is now very much improved – to the satisfaction of the referee, we hope.

Referee #2 (Remarks to the Author):

Summary of the Key Results

The submitted manuscript describes the measurement of magnon modes in a material suitable for such a first attempt: NiO. Dispersion curves of phonon and magnon modes are obtained using a new-generation monochromated scanning transmission electron microscope. The signal resulting from the scattering of a fast electron beam during interaction with these elementary excitations is analyzed with very high energy resolution and collected with a direct detection hybrid detector, maximizing the signal-to-noise ratio and enabling the experimental conditions necessary for detecting the magnonic signal, which has cross-sections several orders of magnitude lower than those of phonon modes. The dispersion curves are acquired under conditions approaching the fundamental diffraction limit, offering the best compromise between spatial resolution and wave vector q -resolution. Thus, this detection is achieved with nanometric spatial resolution, which is undeniably useful for future experiments aiming to probe magnetic inhomogeneities at these scales. The experimental results are corroborated by numerical simulations based on a well-suited theoretical approach.

Originality and Significance

The results obtained are a remarkable experimental achievement. They are convincing and significant for the perspectives they open in the field of magnonics. This is a groundbreaking experiment that validates recent theoretical predictions regarding the possibility of probing magnon modes with a fast electron beam. Following the initial demonstrations on phonons about a decade ago, which are now widely detected globally in electron microscopy down to atomic resolution, this work represents an important milestone in the quest to probe elementary excitations at “low energy” (typically in the THz frequency range) with high spatial resolution via inelastic scattering of fast electrons in an electron microscope. Here, a (purely) magnetic excitation is probed, which inherently exhibits very low intrinsic coupling with a fast electron beam, typically a privileged probe for excitations mediated by Coulomb interactions. This major result has been made possible by recent technological advancements in electron optics and detection sensitivity, positioning EELS as an advantageous alternative to “traditional” techniques for detecting magnetic excitations, such as inelastic neutron scattering. The perceived advantages include multiple orders of magnitude improvement in spatial resolution for probing magnetic responses and structures at the nanometer scale, requiring smaller sample quantities and significantly reduced acquisition times, and a much more versatile experimental setup (with possibilities for multimodal approaches within the same instrument: imaging, diffraction, elemental and chemical analysis, measurement of phonon modes at the nanometric or even atomic scale, etc.). The potential is even more promising for exploring related excitations (e.g., electromagnons or polarons), for which the expected cross-sections can be significantly higher.

Data and Methodology: Validity of Approach, Data Quality, Quality of Presentation

At this stage, this work remains a proof of concept on a material with favorable characteristics, including minimal spectral overlap between phonon and magnon bands. This is an evidently wise choice. The demonstration of the possibility of isolating the purely magnetic excitation signatures from any phononic or hybrid excitation remains to be achieved in the future. The core of the demonstration relies on the acquisition and analysis of phononic and magnonic dispersion curves.

We thank the referee for such a thorough and appreciative summary of our work. We are absolutely delighted that our proof-of-concept experiments were deemed of interest and that the methodology and approach we used were able to not only convince the referee of the validity of our work but also to convey our enthusiasm for what we think is a very exciting development.

Some corrections and additional information could improve the manuscript's presentation and clarity.

[1] Figure 2b: The superimposed yellow lines obscure the experimental data, preventing clear visualization. This should be improved by opting for a better graphical choice (e.g., a dotted and thinner line).

We agree with the referee about the poor legibility of this figure, an issue also raised by referee #1 in their questions 4 and 5. We have now updated Figure 2 with background subtracted ω - q maps in the magnon loss range, which we hope illustrate the magnon dispersion curves more clearly.

The original figure 2 is still presented for completeness in a new Supplementary document, wherein the overlaid yellow lines have been removed.

For the reviewer's convenience, we reprise below a part of the answer already provided in response to reviewer 1's comments, but we refer them to the full arguments to our answers to questions 4, 5 and 6 above.

*The variation of the magnon signal in the experimental data along the momentum axis is now illustrated more clearly in new and updated panels in Figure 2, now labelled Figure 2c and 2d. These display the energy range in the ω - q map corresponding to magnon losses only, where in order to better separate the magnon signal, the background from the decaying tail of the phonon peaks at lower energy losses was removed. A first order log polynomial was used in this instance, a background model often used in the low loss region due to its slower decay tail (see, e.g., R. Egerton, *Electron Energy Loss Spectroscopy in the Electron Microscope*, Springer, 2011). More comments on background subtraction are provided in response to referee #1's question 11.*

The correspondence with simulated data is now far more clearly visible. Taking the example of the 220 data, two magnon dispersion lobes of spectral intensity are clearly visible on either side of the M point, with a maximum peak at 100meV. The lobes' intensity is asymmetric on either side of the M point, with stronger signal in the first BZ (between Γ and M) compared to the lobe between M and Γ' . The intensity tends to 0 towards each of the BZ vertices (Γ , M and Γ').

Author actions (see also response to question 4 by reviewer 1 above)

- Figure 2 has been replaced with an updated version, with fewer overlaid plots, to improve readability. Panels c,d now include background-subtracted ω - q maps instead, which better demonstrate the variation of the magnon signal in momentum space.
- The associated section of the manuscript (lines 109-132) has been expanded to reflect figure changes and describe the magnon dispersion behaviour.
- The original figure is now included as Supplementary Fig. 1 to illustrate the variation of the magnon signal along the momentum axis through the E^2 scaling of the ω - q map.

[2] Wave Vector Integration Range: The specific choice of the wave vector integration range for visualizing the partial magnonic DOS presented in Figures 2c and 2f does not seem to be justified in

the text. Could the authors comment on this choice? How does these DOS features compare to a more broadly integrated DOS (while obviously avoiding regions near the Gamma points)?

The momentum integration range was chosen so as to be centred at the momentum value at which the maximum intensity feature of the magnon signal appears, as predicted by our theoretical simulations and confirmed from reference inelastic neutron scattering signal in Ref 37 [Sun, Q. et al. PNAS 119, e2120553119 (2022)]. This maximum is different along the two directions explored in momentum space, and as discussed in the manuscript, it is comparatively closer to X than to M (in relative value to the size of the first Brillouin zone) in the corresponding directions.

The integration window was chosen as half of the experimental momentum resolution to improve signal by summing over the most intense features, but narrow enough to avoid smearing the data through momentum-averaging. Choosing a much broader window would result in broader averaging (and thus to a broadening of the peak), but also in additional noise in the peak tail, as after background subtraction there is almost only noise left, thereby contributing "almost solely additional noise". A sentence was added to the text to motivate more clearly this choice.

Finally, we note that the spatially resolved experimental data, included in response to referee #1's comments, was acquired with a large convergence angle in dark-field EELS mode. The spectra presented in the new Figure 4 thus correspond to the case highlighted by referee #2 here – of a wide momentum window integration away from Γ . Thanks to the larger collection angular range in this geometry, noise is kept at acceptable levels.

Author actions:

- The Main text has been amended (lines 127-130) to motivate the choice of momentum space range for signal integration presented in Fig.2 e,f.
- Extended Data Fig. 4 has been introduced to facilitate comparison with theoretical calculations of magnon EELS.

[3] Simulation Approach: The authors use simulations based on a semiclassical approach derived from a standard multislice method, commonly applied when Coulomb interactions dominate. Although the technical details of this approach have been already published, it would be helpful here to describe, without delving into formalism, the physical (and not technical) elements included in the simulations that support the attribution of a magnonic signature to the measured experimental signal.

We thank the referee for this suggestion, which we agree would provide a useful reference for prospective readers of our work. The theoretical methodology section has been expanded accordingly, to include a short physical description of both the frequency-resolved frozen phonon multislice method (FRFPMS, used to calculate phonon excitations), and of a newly developed calculation framework called the time-autocorrelation of auxiliary wavefunctions (TACAW), which we deployed here to calculate magnon spectra and dispersion diagrams. The full formalism for the latter has now been published (our original submission contained a reference to the online preprint of this work), and the corresponding reference has been updated.

Author actions:

- The Methods section (lines 494-515) has been expanded to describe the time-autocorrelation of auxiliary wavefunctions (TACAW) calculations framework used in this work and reference [35] has been updated to cite the now published paper fully detailing the methodology.

[4] Comparison of Experimental and Simulated DOS: It would be useful to compare the experimental and simulated partial DOS. Could the authors provide this additional information?

We very much agree with the reviewer. To aid the comparison between theory and experiment we have now included a new Extended Data Figure 4, where the experimental background subtracted magnon ω - q maps, as also shown in Figures 2 c and f, are displayed side-by-side with the corresponding calculated magnon EELS maps.

Additionally, experimental spectra, integrated across the same wave-vector window as in Figures 2e and 2f (see our response to the reviewer's question about wave-vector integration above), are also displayed side-by-side with corresponding simulated spectra with the same integration range.

As a side comment, we note that our calculations provide momentum-resolved electron energy loss spectra (rather than densities of states), and are thus directly comparable to experimental data obtained in the electron microscope.

Author action:

- New Extended Data Figure 4 has been introduced to aid comparison between theory and experiment, for both dispersion curves and integrated spectra.

[5] References : the authors provide a comprehensive and appropriate review of the use of particle beams to probe magnetic excitations, highlighting the benefits of EELS in terms of spatial resolution improvement. It would be beneficial to expand this review to include spatially-resolved techniques and mention the implementation of vector magnetometry, which is an essential tool in characterizing the distribution of currents and magnetization across a wide range of systems. Point defect sensors, like the nitrogen vacancy (NV) center in diamond for example, have shown impressive sensitivity and spatial resolution for detecting these fields, and are crucial for characterizing magnetic textures such as skyrmions or cycloids with varied morphologies, as well as imaging complex antiferromagnetic orders at the nanoscale.

We thank the reviewer for bringing to our attention this field and its applications, which we agree would be a very relevant addition to our literature review. Due to space constraints, we are only able to provide a rather succinct mention of this area of research, together with a couple of additional references, which we hope will point the interested reader towards more in-depth discussions. Should the reviewer feel that there are more appropriate references, or work of a seminal nature in this field that we overlooked, we would be happy to further add these to our literature review.

Author action:

- The manuscript was modified to mention these techniques (lines 40-40) and additional references [18, 19] were introduced in the text.

Referee #3 (Remarks to the Author):

Detecting magnons using TEM-EELS is a valuable approach for investigating the spin dynamics and magnetic properties of materials. However, a key challenge lies in the relatively weak inelastic scattering cross-section of magnons compared to phonons, as both types of excitations typically occupy the same energy range in most materials. In this study, the authors demonstrate the detection of a magnon band at approximately 100 meV in an NiO sample, with the assignment corroborated by simulations. This is a challenging experiment with intriguing results, possibly give high energy magnon dispersions with spatial resolution, beyond the inelastic neutron scattering approach. However, the broader significance of this work needs to be better clarified.

[1] Firstly, the separation of magnon and phonon signals in NiO is achievable due to their specific energy difference and the E^2 amplification factor. To strengthen the impact, it would be beneficial to discuss how this method might apply to other materials or heterostructures with less pronounced energy separations.

We thank the referee for this comment regarding the wider applicability of magnon-EELS in the STEM given the experimental acquisition difficulties, a point similarly raised by referee #1.

For this proof-of-principle demonstration, we focused on a favourable materials system for which the magnon losses are well separated in energy and momentum from other spectral contributions – a choice we note referee #2 highlighted as “wise”. Our manuscript also clearly highlighted the challenges of these and of future experiments, in particular the intrinsically very weak signal levels, while offering possible mitigation strategies.

Nevertheless, we agree with the need to better highlight this discussion and the implications for further and wider deployments of the technique in the future, which we hope to have achieved through revisions in the text.

Paraphrasing our response to reviewers #1 and #2, the ability to increase detection time thanks to hardware improvements (optimised detectors, stable stages) without loss of spatial resolution, as well as the use of additional experimental levers (variable temperature, electrical biasing and MEMS technology) to boost the magnon signal intensity relative to other spectra contributions, will allow us to overcome challenges related to detection efficiency. By extending acquisition times, we can achieve the necessary signal-to-noise ratio for high-quality magnonic data on numerous other systems without compromising spatial resolution.

*Furthermore, many of the materials of interest for magnonics, including **metallic and oxide systems, present favourable separation between signals and higher intrinsic magnon scattering cross-sections**. There are therefore plenty of additional systems to explore as the technique further develops, as we now explicitly discuss in the manuscript.*

*Finally, exploiting the **interaction of magnons with other quasi-particles** whose scattering involves energy losses in the same range can lead to enhanced signals or unique dispersive behaviours due to quantum effects, which could be directly exploited to fingerprint magnon scattering with signal strength presenting less of a limitation. This is further explored in our response to the reviewer’s next question.*

Author actions:

- The discussion has been modified and expanded to highlight and to clarify these possible strategies for mitigation of the low magnon signal (lines 244-272).

[2] Many spintronic applications rely on phonon-magnon coupling in the low-energy regime, where phonon and magnon excitations mix. Therefore, examining cases where the phonon spectrum varies across magnetic phase transitions with high spatial resolution could add further relevance and interest.

We are in perfect agreement with the referee about magnon-phonon coupling effects. Preliminary studies of magnon-phonon polarons are already under way, both by our own group and others around the world – as pointed out in the discussion and through references 44 and 45 in the original manuscript [A. Reifsneider et al., Microscopy and Microanalysis 30, ozae044.772 (2024); and K. El Hajraoui et al., Microscopy and Microanalysis 28, 2338–2339 (2022)].

As we write in the manuscript, “these interactions are characterised by modifications of the observed dispersion diagrams (e.g., changes in dispersive behaviour, or the appearance of spectral band anti-crossings), or to the spectral bands’ dependence on temperature. Preliminary reports suggest this strategy, although indirect, may be less demanding on signal sensitivity.”

Similarly, systems containing magnetic phase or domain boundaries across which the magnon response is expected to vary spatially are of the utmost interest. Our newly acquired spatially resolved data, now presented in Figure 4 of the main text, shows how the magnon EELS response is confined to a NiO thin film bounded by a non-magnetic substrate (YSZ) on one side, and vacuum/carbon on the other, illustrating a simple test-case of this spatial variability.

More complex highly localised variations may be detectable through STEM-EELS, but we would also point to the inherent difficulty in interpreting data from such systems, given that the appropriate theory describing electron scattering and providing simulations of EELS spectra arising from such coupling effects, are still being developed. We hope to be able to present relevant results in the future.

Author actions:

- A new Fig. 4 describing spatially resolved magnon dark-field EELS measurements across a NiO thin film has been introduced.
- The associated section in the Main text (lines 192-218) and Discussion (lines 230-235) sections of the manuscript have been expanded to reflect the new figure addition and supporting theoretical calculations.
- New Supplementary note 3 and corresponding Supplementary Fig. 3 outline the theoretical calculations of the spatial dependence of magnon scattering across a NiO slab.
- The discussion of magnon coupling with other quasi-particles has been slightly amended to highlight this as an effective strategy to fingerprint (albeit less directly) magnon scattering.

[3] Figure 2 and its caption lack clarity. The 220 lines appear nearly symmetric around the M point, while the 002 lines are asymmetric. A better explanation is needed for the asymmetry observed in panels b and e; is it an intrinsic effect of momentum-dependent scattering or a result of experimental limitations? Additionally, if the green areas in panels c and f represent integrated contributions, it is unclear why they drop to zero below 70 meV, unlike panels a and d.

We now recognise the lack of clarity in our original figure 2 – a point of criticism raised by all three referees.

We have now fully revised figure 2, removing the overlaid line profiles, and introducing new panels showing background-subtracted ω - q maps in the magnon loss range, which we hope illustrate the magnon dispersion curves more clearly. These now highlight the striking match with calculated dispersion magnon curves. A side-by-side comparison of experiment and theory is provided for convenience in Extended Data Figure 4, for both the dispersion diagrams and spectral profiles integrated over a small momentum window corresponding to the magnon signal intensity maximum.

Taking the example of the 220 data, two magnon dispersion lobes of spectral intensity are clearly visible on either side of the M point, with a maximum peak at around 100meV. The lobes' intensity is asymmetric on either side of the M point, with stronger signal in the first BZ (between Γ and M) compared to the lobe between M and Γ' . The intensity tends to 0 towards each of the BZ vertices (Γ , M and Γ'). While the asymmetry is also evident in the 002 data, the separation between 'lobes' is less pronounced. This is not surprising as the Γ -X- Γ' distances are shorter in this direction, with the moderate momentum resolution and relatively high signal to noise ratio making it more challenging to obtain a clear separation between branches.

As noted in our response to referee #1's questions 6 and 7, the magnon intensity's asymmetry is expected from theory, as are differences in intensity between the first and higher order BZ. This already is illustrated to an extent by the intensity asymmetry of the magnon 'lobes' on either side of the X and M points in both experimental and simulated EELS data, with the intensity being stronger in the first BZ. Here, due to geometrical constraints in the experiment and to maintain energy resolution across the entire momentum selecting slit, our experiments do not extend beyond the second BZ (first order Bragg reflection). However, the inelastic neutron scattering data and associated simulations in Reference 37 in the original submission [Sun, Q. et al. PNAS 119, e2120553119 (2022)] explore a wider momentum transfer range. They point to a similar behaviour, with what seems to be a strong dampening of the magnon signal in the higher order BZs.

Author action:

- Figure 2 has been replaced with an updated version, with fewer overlaid plots, to improve readability. Panels c,d now include background-subtracted ω - q maps instead, which better demonstrate the variation of the magnon signal in momentum space.
- The associated section of the manuscript (lines 109-132) has been expanded to reflect figure changes and describe the magnon dispersion behaviour, including the signal asymmetry.
- The original figure is now included as Supplementary Fig. 1 to illustrate the variation of the magnon signal along the momentum axis through the E^2 scaling of the ω - q map.
- A new Extended Data Fig. 4 has been introduced to facilitate comparison with theoretical calculations of magnon EELS.

[4a] The reliability of subtracting the power-law background from the LO phonon signal to extract the weak magnon signal should be addressed.

The impact of background model choice is discussed in detail in our response to referee #1 above. To re-iterate here for the referee's convenience, the use of a power-law function to remove any decaying background is one of the most commonly used approach in EELS, as it models population statistics quite effectively and appears to be relatively robust, even in the vibrational EELS range [e.g., K.L.Y. Fung et al., Ultramicroscopy 217, 113052 (2020) or recently Haas B, et al., Nano Lett. 2023, 23, 13, 5975–5980]. However, the stability of the model across neighbouring pixels in 2-dimensional datasets can be poor

when the energy fitting window is narrow (for very low energy losses close to the zero-loss peak tail, or for signals superimposed on the tail of other losses close in energy). This can lead to subjective results depending on a given user's fitting window selection. As a result, most research groups in the field employ background subtraction on a case-by-case basis, considering various background options, which can sacrifice physical justification to improve visibility [Hachtel JA, et al., Sci Rep. 2018, 8 (1), 5637].

Here, we initially chose to present our magnon data with minimal user intervention. The scaling by the square of the energy loss is entirely user-agnostic, and while it has some physical justification (especially at very low energy losses, see Supplementary note 1), it can be thought of as a pure 'data scaling' strategy (as would a logarithmic intensity display be), helping to visualise on the same panel signals with vastly different intensity levels.

We had only used power-law subtracted spectra in a number of figures as an illustration of the subtle magnon feature superimposed on the tail of the LO phonon.

However, as the consensus among reviewers indicated that the presentation of the magnon dispersion curves could be improved, we explored different fitting options for the removal of the LO phonon decaying background immediately before the energy range where our simulations predict the presence of the magnon signal. A new Supplementary Figure 2 presents the ω - q maps along the 220 systematic rows of reflections, using first-order logarithmic polynomial and power-law models for comparable energy windows.

In both cases, the resulting signal shows a remarkable qualitative resemblance to the calculated curves in Figure 3 of the main manuscript. However, the use of the power-law background appears to yield noisier maps, as it is more susceptible to variations in phonon intensity tail across the narrow fitting energy window, resulting in some negative values, particularly on the weaker M-F' magnon branch. Therefore, we have decided to retain the first-order logarithmic polynomial subtracted data for a new, update figure 2 in the main manuscript, and have updated the spectra in Figures 2e and 2f accordingly.

For completeness, the full ω - q maps, displaying the entire energy range and including the zero-loss line, phonon and magnon branches, both without scaling and with scaling with the square of the energy loss, as initially shown in our original figure 2, are also still presented for completeness as supplementary figure 1.

Author action:

- Figure 2 has been updated to include background subtracted ω - q maps (c,d) and spectra (e,f) produced using a log-polynomial background fitting method to model the decaying phonon intensity.
- Supplementary note 2 and corresponding Supplementary Figure 2 have been introduced to discuss the robustness of background subtraction

[4b] [...] the potential for shifting the arrowed regions leftward to emphasize stronger signals [should be addressed].

The momentum integration range and position, highlighted by the arrows, were chosen so as to be centred at the momentum value at which the maximum intensity feature of the magnon signal appears, as predicted by our theoretical simulations and confirmed from reference inelastic neutron scattering signal in Ref 37 [Sun, Q. et al. PNAS 119, e2120553119 (2022)]. This maximum is different

along the two directions explored in momentum space, and as discussed in the manuscript, it is comparatively closer to X than to M (in relative value to the size of the first Brillouin zone) in the corresponding directions.

The integration window was chosen as a fraction of the experimental momentum resolution to improve signal by summing over the most intense features, but narrow enough to avoid smearing the data through momentum-averaging. Choosing a much broader window would result in broader averaging (and thus to a broadening of the peak), but also in additional noise in the peak tail due, as only noise is left after background subtraction away from the area of maximum intensity. A sentence was added to the text to motivate more clearly this choice.

Author action:

- The Main text has been amended (lines 131 -133) to motivate the choice of momentum space range for signal integration presented in Fig.2 e,f.
- Extended Data Fig. 4 has been introduced to facilitate comparison with theoretical calculations of magnon EELS.

[5] The agreement between theory and experiment could be improved, as discrepancies are noticeable: the spectra in Fig. 3b and 3d differ from those in Fig. 2b and 2e. It would be helpful to present the magnon bands, equations of electron-magnon scattering cross-sections, and perhaps an animation of the 100 meV magnon mode to better illustrate theoretical advancements and support reader comprehension.

We hope that the significant changes made to figure 2, and the inclusion of a side-by-side comparison of theoretical and experimental magnon dispersions and integrated spectra in Extended Data Figure 4 now illustrate what we feel is an excellent agreement between experiment and theory.

An additional description of the physical principles underlying the simulation results has been added to the Methods section, in particular concerning the recently developed Time-Autocorrelation of Auxiliary Wavefunctions (TACAW) method. While expanding on the full formalism would be beyond the scope of this manuscript, the full algebraic treatment of this theoretical framework is available elsewhere. The reference to this work has been updated (we originally referred to an online pre-print – the paper has now been fully published).

Overall, given the technique and the theoretical calculation framework are in their infancy, this being the first demonstration of magnon spectroscopy in the STEM, we believe our results offer remarkably little discrepancy. It goes without saying, however, that as the technique matures we hope to improve on these initial steps – as has been the case with vibrational spectroscopy in the STEM in the 10 years since its demonstration.

Author action:

- Figure 2 has been replaced with an updated version, with fewer overlaid plots, to improve readability.
- Extended Data Fig. 4 has been introduced to facilitate comparison with theoretical calculations of magnon EELS.

Response to Reviewers – Manuscript 2024-10-21687A

Referee #1 (Remarks to the Author):

I would like to thank the authors for their nice work in addressing the review comments, particularly regarding my primary concerns about the robustness and reliability of the “signal”. The addition of the new Figure 4 [...] significantly strengthens the paper [...], providing spatial evidence to support the identification of these signals as magnon-related. Other improvements to the manuscript have also enhanced its overall quality. Overall, the robust demonstration of magnon dispersion measurement represents a significant breakthrough in the fields of physics and electron microscopy.

We are delighted to read that our response satisfactorily addressed the referee’s comments, and that they feel that the manuscript has been strengthened “significantly” to the point that our “demonstration of magnon dispersion represents a significant breakthrough in the fields of physics and electron microscopy”.

However, some of my questions about how surfaces and interfaces influence magnon behavior remain unanswered. This limitation may stem from the current signal-to-noise ratio constraints caused by the small scattering cross-section of magnons in the EELS experiments. While these scientific questions could be explored in future studies, they do not diminish the value of this work, which focuses on methodological demonstration and conceptual validation.

We also share the view that there are many more exciting avenues to explore and more experiments to carry out now that magnon detection in the STEM has been achieved. As the referee suggests, this can and should be done in future studies “without diminishing the value” of the present work.

My remaining concern lies in strengthening the evidence for distinguishing magnon signals from other possibilities, especially given the notable discrepancies between simulations (Fig. 3b,d) and experimental data (Fig. 2c,d). While the intensity variations in Fig. 2c show partial agreement (e.g., low intensity at the M-point), the strong intensity at the X-point in Fig. 2d contradicts Fig. 3d, which the authors attribute to insufficient momentum resolution. The energy discrepancies between simulations and experiments (peaks between Γ -M/X in simulations versus no clear change along Γ -M- Γ and monotonic change along Γ -X- Γ in experiments) also require clarification. I wonder whether these phenomena are reproducible and reliable.

We agree with the referee that the quantitative correspondence between experimental data and numerical simulations is not perfect, but we would respectfully suggest that the description of “notable discrepancies” is overly harsh to what we feel is a very convincing qualitative match overall.

While we cannot rule out other sources, we suggest that the limited momentum and energy resolutions are the most likely explanations for the imperfect match.

To better demonstrate the effect of finite momentum and energy resolution, we are incorporating an additional Supplementary Figure 4, wherein the calculated dispersion diagrams for both $\Gamma \rightarrow M$ and $\Gamma \rightarrow X$ directions have been broadened to match experimental limitations, convolving with a top-hat function of 0.7 \AA^{-1} width (mimicking the momentum-selecting slit) and with a Gaussian of 11 meV full-width at half-maximum (reflecting the effective energy resolution through the sample after integration of 90000 frames). This clearly shows how the magnon peaks in the tighter Γ -X- Γ' direction appear to merge into a more continuous lobe of intensity, reflecting the experimental data where, as discussed in the main

manuscript, the two peaks are hard to separate on either side of the X point. The visual match in the Γ -M- Γ' direction, where the Brillouin zone vertices are spaced further apart so the magnon lobes are well resolved, is also more 'pleasing' (albeit less defined) after this forward convolution.

Furthermore, and also in response to the referee's point 3 below, we are now also showing additional data with fewer integrated frames than used to generate the main manuscript's Figure 2, presented in a new Supplementary Figure 3. The shorter total integration time results in noisier signal, but the signal broadening/blurring is lessened – so that for the Γ -X- Γ' direction this reveals how the two lobes of intensity on either side of the X point are indeed also resolved, with the magnon signal dropping to zero at the X point, and with an asymmetric intensity on either side of X, an effect predicted by the simulations.

Finally, wish to emphasise that magnon-EELS simulations are still in their infancy, with theoretical and numerical frameworks—developed in part by our group—actively evolving. At present, these approaches do not yet incorporate magnon-phonon coupling or fully capture the complex influence of phononic backgrounds. A fully coupled magnon-phonon framework, which is not yet available, would be required to reproduce the complete spectral landscape, including the intricate interplay between signal and background, as well as to address effects such as background subtraction, noise, and broadening in a more systematic and quantitative way. As such, the current calculations should not be viewed as an “absolute truth” to which experiments must precisely conform. Nonetheless, the uncoupled simulations presented here enable a clear identification of magnonic features based on their dispersion behaviour and alignment with Brillouin zone boundaries and high-symmetry points—providing compelling evidence for the interpretation of the experimentally observed excitations. These theoretical developments are ongoing and will be the subject of future communications and publications.

Author actions:

- New Supplementary Fig. 4 shows the effect of broadening on the calculated dispersion diagrams to reflect the limited energy and momentum resolution in experiments, improving the visual match between theory and experiment.
- New Supplementary Fig. 3 shows clear separation of the magnon lobes in the Γ -X- Γ' direction when fewer intensity frames are averaged, albeit at the cost of overall noisier signal. This further points to limited resolution (in both energy and momentum) as the main reason for the imperfect match between experiments and simulations.
- Additional commentary in Supplementary note 4 highlighting the early stage of development of the theoretical framework for magnon-EELS calculations and on-going work.

1. In fact, I was expecting that authors have performed the spatial mapping similar to Figure 4 but with a smaller convergence angle (using a slit aperture rather than the round one), which could reveal how the signals distribute in space across different momentum points (e.g., Γ /M/X-points). Current limitations in momentum resolution (due to the large convergence semi-angle ~ 31 mrad in dark-field EELS) prevent such analysis.

We agree with the referee that spatially resolved 'maps' of dispersion diagrams, especially across defects or hetero-interfaces are a particularly exciting prospect for this technique.

The demonstration of such experiments in the case of phonon spectroscopy (R. Qi et al., Nature 599, 399-403, 2021) was however only possible after almost 8 years of methodological developments and

technical improvements following the initial report of vibrational spectroscopy in the STEM (O.L. Krivanek et al., Nature 514, 209-212, 2014), and 4 years after our own group demonstrated nanoscale momentum-resolved vibrational spectroscopy (F.S. Hage et al., Science Advances 4, eaar7495, 2018).

We look forward to following in these footsteps and to exploring the spatial variations of magnon dispersions across defects and interfaces in future work, taking advantage of the on-going developments, which we highlight in the main manuscript discussion, especially in terms of signal strength optimisation to make these experiments easier.

Nevertheless, we believe that a slot aperture and small convergence angle would be the wrong choice of scattering geometry for the systems explored in our present work and for the intended purpose of our spatially resolved experiments, that is, highlighting the spatial localisation of the magnon signal and further confirming its origin as magnon scattering.

To further demonstrate the pertinence of our choice of a large convergence angle, we have added a new Supplementary Figure 6 that offers more insights into the spectral variations detected across the YSZ/NiO and NiO/vacuum interfaces, only observable using a high convergence angle. These further strengthen the interpretation of the signal as arising from magnon scattering.

The interface between YSZ and our NiO thin film is atomically sharp, and the selection of a large convergence angle and of the dark-field EELS geometry ensures the highest possible resolution across the interface, which a small convergence angle together with a slit EELS aperture would preclude (the spatial resolution would be $\sim 1.5\text{nm}$, as in our own momentum-resolved experiments, resulting in signal mixing).

This geometry allows us to observe a localised broadening and subtle redshift of the main magnon mode at 100meV in the vicinity of the thin film surfaces, while still inside the film (a position that can be maintained precisely thanks to the small 0.1nm probe and large 31mrad convergence angle), together with an intensity dampening of the magnon signal. While probe propagation could still play a role in the observed signal changes, as we mention in the main text in our discussion of Fig. 4, magnon EELS simulations for a thin slab of NiO also predict a similar effect in the vicinity of the slab's surfaces (Supplementary Fig. 7 and Supplementary Fig. 8).

Although our NiO thin film is still "thick", we believe this broadening is related to the appearance of confinement-related additional, softer magnon modes in the ultra-thin film limit (J.A. do Nascimento et al., Phys. Rev. B 110, 024410, 2024), a phenomenon we plan to study in further detail in future work.

For example, if the "signal" is magnon-related, stronger intensity should appear in NiO regions when the momentum is between Γ and M, while weaker signals would occur at Γ and M points based on experimental data Fig.2c and simulated data Fig.3d. These momentum-resolved spatial mapping could also clarify whether discrepancies arise from experimental or simulation limitations.

YSZ does not sustain magnons, but exhibits a main optical phonon band around 76meV , which we observe in our spectra. Spatially resolved dispersion diagrams on either side of the interface would simply show the individual contributions of YSZ (phonons only) or NiO (phonons and magnon, as we already show), with no more information regarding the origin of the spectral signal than already obtained from our existing momentum-resolved experimental data. Namely, the 100meV magnon signal is already weaker at Γ and M within the NiO, and would simply be absent within the YSZ substrate across momentum space.

At the same time the larger probe size resulting from the small convergence angle, as suggested by the referee, would result in spectral overlap close to the interface, making disentangling the origins of the observed signal even more difficult.

Finally, we note that possible interactions at the interface are unknown (and, arguably, not expected in this model system), while the exploration of magnon interfacial effects in relevant heterointerfaces would necessitate extensive theoretical developments far beyond the scope of this work.

Author actions:

- Supplementary Figure 6 has been added to highlight spectral fine structure changes localised to the surface and interface of the NiO thin film, in agreement with theoretical calculations, emphasising the pertinence of our choice of experimental geometry, with a large convergence angle.
- Supplementary Note 6 details these observations, and describes the fully separated and distinct spectral signal observed in the NiO, YSZ (and in a reference carbon area, in response to the referee's third question, below).
- The discussion in Supplementary note 6 suggests future work on spatially resolved dispersion diagram maps across structural features and defects, with an additional reference to relevant literature (R. Qi et al., Nature 599, 399-403, 2021).

2. In Extended Data Figure 4, presenting spectra from additional momentum windows (e.g., X/M-points) alongside existing data would provide intuitive evidence of magnon dispersion. Direct comparisons between these spectra could strengthen the interpretation.

In order to maintain optimal energy resolution along the entire length of the momentum-selecting slit, and to ensure good momentum sensitivity along the length of the slit by spreading out the row of reflections as much as possible, we chose necessarily to limit the number of Brillouin zone vertices observed across a full camera frame, limiting the width of the momentum windows we can present.

Nevertheless, in response to this suggestion (and also in response to the referee's question 3 about repeatability), we have now added a new Supplementary Figure 5, which shows an additional, independent dataset with a wider momentum window, with symmetric data on either side of Γ . Due to the wider angular (momentum) range, some asymmetric energy resolution loss due to spectrometer aberrations results in some minor feature blurring on the Γ -M Γ'' direction, compared to Γ -M Γ' : the phonon bands are not as well resolved, and the magnon lobes not as clearly separated on the right-hand-side of the Γ point. However, the phonon and magnon bands are clearly recognisable on both sides.

Author actions (see also response to point 3 below):

- A new Supplementary Fig. 5 shows an additional, independent dataset. This was acquired on a different day, by a different operator, and on a different area of the sample, but in otherwise similar conditions. Here a wider momentum window was chosen, but this resulted in minor loss of energy resolution across the slit due to spectrometer aberrations at high angle.
The phonon and magnon bands are clearly recognisable on both sides of the Γ point.

3. Given the overlapping energy ranges between magnons and amorphous carbon vibrations, authors may need to discuss and exclude this possibility.

We thank the referee for suggesting this clarification. For completeness, we now show in Supplementary Figure 6d a spectrum recorded from an area of carbon protection layer far outside the field of view of the spectral map, covering the same energy range and processed identically to the magnon spectra in the same figure. The FIB sample preparation had resulted in the formation of a hole (vacuum) immediately above the NiO film in the spectrum image region-of-interest, but some remaining carbon material from the protective cap remained present further above the surface in other, extended-range data, allowing for a direct and self-consistent comparison.

This ‘carbon spectrum’ can be directly compared to the spectra from YSZ and NiO, showing a completely different spectral shape and spectral separation, and allowing us to unambiguously exclude carbon from contributing to the data assigned to the magnon bands.

We also note that in the ultra-high-vacuum conditions used in our instrument (the vacuum at the sample is typically $<9 \times 10^{-10}$ Torr), adventitious carbon build-up (contamination) is hardly ever observed, even after hours-long observation. Some extremely faint intensity at >150 meV, i.e. in the range corresponding to carbon, and which could have arisen from some minor contamination build-up, can be faintly seen in some of the spectra in Supplementary Fig. 6c and d, in the ‘surface’ spectrum (some carbon may remain on the very surface), and in the ‘bulk’ spectra. Once again, these are distinct in energy range and spectral shape, and of such low intensity compared to any other contribution, that carbon can be excluded with confidence as a spurious contribution to the magnon signal.

Author actions:

- **New Supplementary Figure 6 now compares a spectrum from a carbon protection layer, with the NiO and YSZ spectra, all processed identically. This shows how any carbon spurious contribution to the spectral intensity would have a different energy and shape to the signal attributed to magnon scattering.**
- **Supplementary Note 6 was expanded to include these explanations.**

Moreover, discussing reproducibility is also needed. In fact, through repeated experiments and stacking multiple datasets would improve confidence in the validity of magnon dispersion. With such improvements, the energy variations in the dispersion curves may show better consistence between experiments and simulations.

We note that the data presented already comprises averages of multiple EELS detector frame stacks. For instance (see main manuscript, Methods), Figure 2a is the sum of 90,000 frames, from 6 different and independent datasets of 15,000 frames each, acquired through a period of a day of operation (corresponding to 2 hours of pure acquisition time). This acquisition involved some minor adjustments of spectrometer optics and alignment – but no physical move away from the sample position. Datasets were not averaged over several days, or over several series acquired at slightly different locations (or other changes) to maintain our main goal of ensuring nanometre scale resolution.

Averaging from different areas with different thickness may also influence the observed dispersion, and we feel should be avoided, a main reason why we restrict ourselves to presenting stack averages of up to 90,000 frames only.

To address/illustrate the issue of data accumulation, we now present a new Supplementary Figure 3 and associated Supplementary Note 3, showing how averaging a smaller number of frames (30,000 vs. 90,000 in the Γ -M direction; 15,000 vs. 60,000 in the Γ -X direction) affects the signal.

While noisier, the shorter frame averages do not suffer as much from spectral blurring (due to unintended changes in optical conditions during the hours-long acquisitions). As a result, the magnon dispersion bands are better resolved, in particular in the Γ -X direction (see our response to question 2 above).

Finally, we also present in a new Supplementary Figure 5 and associated Supplementary Note 5 a completely different momentum-resolved dataset, acquired by a different operator, on a different day, and a different area of the sample. This shows a wider momentum window, with symmetric data on either side of Γ . Due to the wider angular (momentum) range in Supplementary Figure 5, some asymmetric energy resolution loss due to spectrometer aberrations results in minor feature blurring in the Γ -M- Γ' direction, compared to Γ -M- Γ'' : the phonon bands are not as well resolved, and the magnon lobes not as clearly separated on the right-hand-side of the Γ point. However, the phonon and magnon bands are clearly recognisable on both sides, demonstrating the reproducibility of the results.

Author actions:

- A new Supplementary Figure 3 and associated Supplementary Note 3 illustrate the effect of data averaging and accumulation.
- An independent dataset in Supplementary Figure 5 illustrates the reproducibility of the signal, as well as its detection across a wider set of Brillouin zone vertices.

Referee #2 (Remarks to the Author):

The version resubmitted by the authors is significantly improved both in terms of presentation and scientific content. The work presented is enhanced by a new high-spatial resolution mapping experiment of magnon modes in a thin NiO layer. The manuscript now combines two remarkable types of experiments: spatially-resolved and momentum-resolved measurements. The responses and modifications made to my questions and suggestions are particularly relevant.

In conclusion, the work presented is of very high quality and great importance; it represents a significant milestone in the field of magnonics.

We thank the referee for their very positive assessment of our work, and for the constructive and helpful comments and suggestions through the review process, which helped us improve and strengthen the presentation of our results.

Referee #3 (Remarks to the Author):

The author provided comprehensive responses to my previous questions, making the paper more suitable for consideration by Nature.

We are delighted that our revisions satisfactorily addressed all the reviewer's previous questions, making in their view our manuscript suitable for publication in Nature.

However, my main concern remains: the TEM-EELS technique, with its poor energy resolution, does not offer a clear advantage for probing magnons. While its spatial resolution is arguably its only advantage over neutron probes, it does not provide insight into local variations in exchange interactions or magnetic anisotropy at surface or interface.

We respectfully disagree with the referee on this point. The energy resolution of STEM-EELS instrumentation (with a current best of approximately 3 meV at 60kV acceleration voltage – see, e.g., N. Dellby et al., Microsc. Microanal. 28, 2640, 2022; and in our experimental conditions: 7 meV) is undeniably poorer than that of techniques such as resonant inelastic X-ray scattering, which can reach μeV energy resolution.

However, for typical magnon experiments in inelastic neutron scattering (INS), the energy resolution is a few % of the incident energy I_0 , and thus of the order of a few meV (see e.g. Q. Sun et al., PNAS 119, e2120553119, 2022) – in other words, very comparable to STEM-EELS.

Furthermore, in addition to the vastly superior spatial resolution of STEM-EELS, which the referee acknowledges and which we demonstrate here, advantages include, among many others:

- *Far shorter acquisition times compared to other inelastic scattering techniques (here, hours, compared to days).*
- *The ability to probe buried features, interfaces and individual nano-objects.*
- *No sample size limitations. INS experiments require large sample volumes (an issue entirely distinct from spatial resolution), sometimes up to gram-levels of material. This places huge demands on sample synthesis and crystal quality, which the use of STEM-EELS entirely circumvents.*
- *The complementarity in a single instrument of all other analytical tools, such as atomic-resolution imaging, chemical mapping etc... allowing direct correlative characterisation.*
- *Wide-band spectral acquisition. While the combination of detector size and energy dispersion places some constraints on the simultaneously achievable spectral range and resolution, STEM-EELS is intrinsically a wide-band technique, probing excitations from the meV to the keV range with the same probe and experimental set-up.*

Finally, there is no physical or fundamental limitation to the ability of STEM-EELS to derive exchange interactions from magnon data, or to observe magnetic anisotropy at interfaces.

Just as exchange interaction parameters (J_{ij}) are obtained from INS data through fitting of the magnon bands, the same mathematical procedure would be possible from our magnon-EELS data. Of note, any observed variation in magnon dispersions at surfaces or interface will allow us to determine experimentally the surface J_{ij} s, which is not currently possible with any other technique. Furthermore, one will be able to infer the strength of the interactions by possible relative softening or hardening of

the magnon modes between the bulk and the surfaces (or at interfaces), offering additional information compared to bulk/surface techniques.

While extremely high momentum resolution is necessary to obtain accurate exchange interaction parameters beyond first neighbour pairs, this challenge is the same for INS – and thus we feel STEM-EELS is not at a disadvantage.

Here, we chose to present a proof-of-principle demonstration of the detection of magnons (and their dispersions) in EELS on a nanometre sized NiO crystal, so that fitting our data to obtain the J_{ij} s appeared unnecessary, and we chose instead literature values for the exchange interaction parameters needed for our numerical simulations.

Author action:

- The wording of the introduction was amended to further highlight the benefits and complementarity of the STEM-EELS approach compared to bulk/surface techniques for magnon spectroscopy.
- The wording of the discussion and conclusion was amended to point to the ability of the STEM-EELS approach to enable the extraction of exchange interaction parameters and possible anisotropy effects by simply applying the same numerical approaches widely applied in the field to the local data obtained in the STEM. A short note on this topic was also added in Supplementary Note 6: Spatially resolved data.

The absence of magnons in YSZ is unsurprising.

We agree – and this was indeed the point of the experiment. Had some spurious signal been observed in the same energy-loss window in the YSZ substrate, it would have invalidated our assignment of the observed peaks to magnon scattering. This observation is an important additional argument in our analysis.

More importantly, the fact that the magnon signal disappears within a sub-nanometre distance of the NiO thin film demonstrates that magnon-EELS signals can be localised and detected at this atomically fine scale – a conclusion that was far from foregone, and which paves the way to further work on near-atomic-level work on magnon scattering, that is, at the lengthscales that are relevant for device physics.

The author should better demonstrate the broader usefulness of these measurements beyond being a first-of-its-kind study.

Here again we will respectfully disagree with the referee. On the one hand, we feel we have robustly motivated our use of a small electron probe to detect magnons, and that we have thoroughly discussed the advantages over, and the complementarity of this approach with, other experimental techniques for magnon spectroscopy (or magnon detection), making it a unique new experimental tool for magnon detection.

More widely, we argue that the very fact that our work is a “first-of-its-kind” study, as the referee recognises, is in itself a very strong argument for consideration here. Numerous proof-of-principle demonstrations of a new experimental technique on model materials systems have appeared in these pages, going on to open new fields of research and creating new research communities. From the field of microscopy and electron energy-loss spectroscopy alone, examples such as vibrational spectroscopy

(O.L. Krivanek et al., Nature 514, 209-212, 2014), electron magnetic circular dichroism (P. Schattschneider et al., Nature 441, 486-488, 2006), or electron vortex beams (J. Verbeeck et al., Nature 467, 301-304, 2010) among many others, illustrate the relevance and impact of such an approach.

We look forward to seeing where the community will take this nascent field and technique of magnon-EELS in the STEM, and to contributing to many other seminal results ourselves – in future upcoming work.

Additionally, the meaning of the green regions in Figures 2e and 2f should be clarified in the caption. Surprisingly, these panels were even not discussed in the text.

We thank the referee for pointing out these omissions. The caption for Figure 2 has been revised to explicitly state that the green shaded areas are the background-subtracted spectra, a visualisation that is widespread in the community. The panels are now referred to in the main text.

Author action:

- The Main text has been amended (lines 131 -133) to explicitly point to the extracted spectra (with signal integration) presented in Fig.2 e,f.
- The caption for Figure 2 has been revised with details on panels e and f.

Response to Reviewers – Manuscript 2024-10-21687B

Referee #1 (Remarks to the Author):

I remain confident that using the small convergence semiangle and slit aperture with improved SNR through technical advancements in future, could fully resolve the previously mentioned signal self-consistency issues and enable studies of surface/interface effects. [...] I do suggest the authors include a discussion or outlook on unresolved questions in the manuscript to highlight the research's open challenges, some of which I have mentioned above.

The submitted manuscript comprises a thorough discussion of possible next technical steps to overcome limitations in signal levels and signal-to-noise intrinsic to the type of scattering giving rise to magnon peaks. This can be found on lines 258 onwards. Similarly, an outlook on materials selection and on-going theoretical developments are discussed at length in lines 238-248 of the main manuscript.

The prospects for spatially-resolved maps of dispersion behaviour across interfaces (using a small convergence angle and an EELS slot aperture) is also discussed in Supplementary Note 6, with a reference to the work by Qi et al. on the topic. To make it even clearer, the text has been amended to explicitly state that this geometry involves the use of a small convergence semi-angle and a slit EELS aperture, to balance spatial and momentum resolution.

Author actions:

- The wording in Supplementary note 6 now explicitly outlines the future use of a small convergence angle and slit EELS aperture for so-called “4D-EELS” experiments, to obtain spatially resolved momentum dispersion mapping.

[...] incorporating statistical error analysis (e.g., standard deviation, consistency across repeated measurements) for key results would strengthen the reliability of their conclusions.

In order to demonstrate the statistical significance of the spectral signature of magnons, we are now showing error bars as confidence bands at a 5σ (standard deviation) level for the two main magnon spectra shown in figure 2. The inset panels in figures 2e,f now illustrate how the signal (green shaded area) is clearly above the noise, the latter being estimated by the $\pm 5\sigma$ error bar displayed as shaded pink bands, assuming signal and noise populations follow a Poisson distribution.

The discussion of frame averaging now includes an estimate of the signal-to-noise ratio (SNR) in the extracted magnon dispersion maps. Here the signal is defined as the mean spectral frame-wide intensity in the magnon energy window after background subtraction, in otherwise unprocessed spectral data. The noise in turn is estimated as the standard deviation of the signal in an energy window beyond any predicted magnon or phonon contribution, beyond 200 meV energy loss. In particular, these values are provided throughout Supplementary Note 3 and in the main manuscript on line 110, as well as in the caption of figure 2. Additional conventions for calculating the SNR, under the assumption of a purely Poisson-distributed noise (valid for counting detectors), are also provided for completeness in an additional Supplementary Table 1.

This illustrates clearly how frame averaging improves the SNR, but that even in the shorter time-averaged datasets, the SNR is substantially above a SNR of 10 providing a clear confirmation of the significance of the recorded data.

Author actions:

- Estimates of the SNR are provided for the various magnon band dispersion diagrams, including for the various levels of frame averaging.
- Confidence bands at a 5σ level provide an estimate of the measurement error and thus of the statistical significance of the data. They are displayed on Figures 2e,f by pink shaded areas, demonstrating unambiguously how the signal is significantly above noise level.